# Optimizing the terrestrial ecosystem gross primary productivity using carbonyl sulfide (COS) within a two-leaf modeling framework

Huajie Zhu[1‡], Xiuli Xing[2‡], Mousong Wu[1*], Weimin Ju[1], Fei Jiang[1,3]

1 International Institute for Earth System Science, Nanjing University, Nanjing, China

5   2 Department of Environmental Science and Engineering, Fudan University, Shanghai, China

Frontiers Science Center for Critical Earth Material Cycling, Nanjing University, Nanjing, China

*Correspondence to*: Mousong Wu (mousongwu@nju.edu.cn)

‡ they contribute equally to this work

**Abstract.** Accurately modeling gross primary productivity (GPP) is of great importance in diagnosing terrestrial carbon-climate feedbacks. Process-based terrestrial ecosystem models are often subject to substantial uncertainties, primarily attributed to inadequately calibrated parameters. Recent attention has identified carbonyl sulfide (COS) as a promising proxy of GPP, due to the close linkage between leaf exchange of COS and carbon dioxide ($CO_2$) through their shared pathway of stomatal diffusion. However, most of the current modeling approaches for COS and $CO_2$ do not explicitly consider the

vegetation structural impacts, i.e., the differences between the sunlit and shaded leaves in COS uptake. This study used ecosystem COS fluxes from 7 sites to optimize GPP estimation across various ecosystems with the Biosphere-atmosphere Exchange Process Simulator (BEPS), which was further developed for simulating the leaf COS uptake under its state-of-the-art two-leaf framework. Our results demonstrated the substantial improvement in GPP simulation across various ecosystems through the data assimilation of COS flux into the two-leaf model, with the ensemble mean of root mean square error (RMSE)

for simulated GPP reduced by 20.16 % to 64.12 %. Notably, we also shed light on the remarkable identifiability of key parameters within the BEPS model, including the maximum carboxylation rate of Rubisco at 25 °C ($V_{cmax25}$), minimum stomatal conductance ($b_{H_2O}$), and leaf nitrogen content ($N_{leaf}$), despite intricate interactions among COS-related parameters. Furthermore, our global sensitivity analysis delineated both shared and disparate sensitivities of COS and GPP to model parameters and suggested the unique treatment of parameters for each site in COS and GPP modeling. In summary, our study

deepened insights into the sensitivity, identifiability, and interactions of parameters related to COS, and showcased the efficacy of COS in reducing uncertainty in GPP simulations.

**Keywords**: carbonyl sulfide, gross primary productivity (GPP), data assimilation, ecosystem modeling, parameter optimization

# 1 Introduction

Over the past five decades, terrestrial ecosystems have been absorbing about 30 % of anthropogenic carbon dioxide ($CO_2$) emissions, playing a crucial role in mitigating climate change (Friedlingstein et al., 2022). Driven by the photosynthesis of terrestrial vegetation, gross primary productivity (GPP) is the largest terrestrial carbon flux and plays an important role in understanding terrestrial carbon-climate feedbacks (Luo, 2007; Wang et al., 2021). However, as the direct observations of GPP using atmospheric $CO_2$ observations are confounded by respiration (Hilton et al., 2017), and the modeling of GPP is affected

by a range of uncertainties such as the poorly calibrated parameters (Macbean et al., 2022), the precise quantification of GPP in terrestrial ecosystems has been a major challenge (Canadell et al., 2000; Yuan et al., 2007).

Carbonyl sulfide (COS) is the most abundant sulfur-containing trace gas in the atmosphere with a lifetime of about 2 years (Montzka et al., 2007; Karu et al., 2023). The tropospheric atmospheric mole fraction of COS is approximately 500 parts per trillion (ppt), exhibiting a typical seasonal amplitude of ~ 100–200 ppt (Montzka et al., 2007; Ma et al., 2021; Hu et al., 2021;

Remaud et al., 2022; Remaud et al., 2023; Ma et al., 2023). In the past decade, COS has emerged as a promising tracer for terrestrial photosynthesis (Stimler et al., 2010; Asaf et al., 2013; Launois et al., 2015; Kooijmans et al., 2019) and stomatal conductance (Commane et al., 2015; Wehr et al., 2017; Sun et al., 2022) as the leaf exchange of COS and carbon dioxide ($CO_2$) are tightly coupled through stomata (Sandoval-Soto et al., 2005; Seibt et al., 2010; Wohlfahrt et al., 2012; Whelan et al., 2018). Unlike $CO_2$, which is emitted back to the atmosphere via leaf respiration (Sun et al., 2022), COS is completely destroyed by a

hydrolysis reaction catalyzed by carbonic anhydrase (Protoschill-Krebs et al., 1996) without back-flux in leaves under normal conditions (Stimler et al., 2010). Consequently, the measurement of COS flux is able to provide a direct and independent way to estimate GPP (Sandoval-Soto et al., 2005; Abadie et al., 2023).

In most of the early studies, GPP was directly estimated by scaling measurement of plant COS uptake with the empirically derived leaf relative uptake (LRU) approach or its extensions that incorporate the effects of temperature, humidity, light and

$CO_2$ concentration on stomatal conductance (Kohonen et al., 2022; Sun et al., 2022; Abadie et al., 2023) because of the simplicity of this approach and the sufficiency of it in many cases (Sandoval-Soto et al., 2005; Whelan et al., 2018). In contrast, the process-based model that mechanistically simulates COS plant uptake by incorporating stomatal transport processes were also developed and widely evaluated (Maignan et al., 2021; Kooijmans et al., 2021). Concurrently, the significance of soil COS exchange has also been recognized, leading to the development of a suite of empirical or mechanistic COS soil exchange

models (Kesselmeier et al., 1999; Berry et al., 2013; Launois et al., 2015; Sun et al., 2015; Ogée et al., 2016; Whelan et al., 2022). Process-based COS plant uptake model and soil exchange models have been integrated into land surface models (LSMs) (Berry et al., 2013; Maignan et al., 2021; Kooijmans et al., 2021). Consequently, by constraining the model parameters of LSMs with COS through data assimilation, not only the model variables like GPP are expected to be improved, but also our understanding of ecosystem processes is expected to be significantly enhanced.

Currently, several studies have been conducted to refine the model parameters of LSMs through assimilating the COS data, and thereby optimizing the modeling of water-carbon fluxes (Chen et al., 2023; Abadie et al., 2023; Zhu et al., 2023). Within

a big-leaf framework, Abadie et al. (2023) demonstrated COS could provide mechanistic constraint on stomatal diffusion, and the joint assimilation of COS and GPP is able to improve the model performance of GPP and latent heat. Ecosystem carbon, water and energy processes are interacting and nonlinear, the changes in one process could induce variations in the other

processes. While COS assimilation has proven effective in constraining COS-related model parameters and optimizing GPP, there remains a gap of systematic understanding of the ability of COS to optimize model parameters from different processes. For example, how effective is the assimilation of COS in reducing model prediction uncertainty of GPP as well as the relevant ecosystem processes in different ecosystems?

Due to the dissimilar illumination conditions, there is the significant variability of leaf photosynthesis between sunlit and

shaded leaves (Chen et al., 1999; Pignon et al., 2017; Wang et al., 2018; Bao et al., 2022). It is now clearly recognized that big-leaf models are conceptually flawed and practically inaccurate and sunlit-shaded leaf stratification is necessary to make accurate canopy-level photosynthesis estimation (Chen et al., 1999; Chen et al., 2012; Luo et al., 2018). Consequently, in the process-based LSM that simulates COS plant uptake and photosynthesis in a coupled manner (Ball et al., 1987; Berry et al., 2013), the application of the two-leaf model shows promise for providing accurate simulation of plant COS uptake. In this

context, we have further explored the capacity of COS to constrain the model parameters of a LSM and to optimize GPP within the two-leaf modeling framework.

Our goal is to address the following questions:

To which parameters is the COS simulation sensitive, and what are the differences in parameter sensitivities between COS and GPP?

How effective is COS assimilation in improving model prediction and reducing prediction uncertainty of GPP?

Which process parameters can be well identified by the assimilation of COS?

How do process parameters interact in COS modeling across diverse ecosystems?

To address these questions, we utilized ecosystem COS flux data to optimize GPP across various ecosystems based on the coupling of COS modeling with the two-leaf based Biosphere-atmosphere Exchange Process Simulator (BEPS). Through

Monte Carlo simulations, we conducted a global parameter sensitivity analysis to explore the sensitivity of COS and GPP simulations to model parameters related not only to photosynthesis but also to water and energy. The interaction and identifiability of these parameters were quantified using Monte Carlo optimized parameter sets. Additionally, the effectiveness of COS in constraining model uncertainty in simulated COS and GPP is evaluated.

## 2 Materials and methods

### 2.1 Model description

#### 2.1.1 BEPS basic model

The BEPS model (Liu et al., 1997; Chen et al., 1999; Chen et al., 2012) used in this study is a process-based diagnostic model driven by remotely sensed vegetation parameters, including leaf area index (LAI), clumping index, and land cover type, as well as meteorological and soil data (Chen et al., 2019). With the coupling among terrestrial carbon, water, and nitrogen cycles (He et al., 2021), it simulates photosynthesis, energy balance, and hydrological and soil biogeochemical processes at hourly time steps (Ju et al., 2006; Liu et al., 2015). For photosynthesis, it stratifies whole canopies into sunlit and shaded leaves and calculates GPP for each group of leaves by scaling Farquhar's leaf biochemical model (Farquhar et al., 1980) up to canopy-level with a temporal and spatial scaling scheme (Chen et al., 1999). In this study, the BEPS model stratifies the soil profile into five layers, and the model implicitly solves the soil water content values for these layers (Ju et al., 2010). Over the last few decades, the BEPS model has been continuously improved and has been used in a wide variety of terrestrial ecosystems (Schwalm et al., 2010; Liu et al., 2015). This study uses the BEPS model that simulates water, carbon and energy processes at hourly interval which enables the detection of diel variations of model variables (Xing et al., 2023).

#### 2.1.2 The two-leaf scheme for GPP and COS modeling

The BEPS model simulates the canopy photosynthesis for the sunlit and shaded leaves separately,

$$\text{GPP} = GPP_{sunlit}LAI_{sunlit} + GPP_{shaded}LAI_{shaded} \tag{1}$$

where $GPP_{sunlit}$ and $GPP_{shaded}$ denote the GPP per unit area for sunlit and shaded leaves, $LAI_{sunlit}$ and $LAI_{shaded}$ represent the LAI values of sunlit and shaded leaves, respectively. $LAI_{sunlit}$ and $LAI_{shaded}$ are calculated as (Chen et al., 1999):

$$LAI_{sunlit} = 2\, cos\,\theta \left( 1 - e^{\left(1 - \frac{0.5\Omega LAI_{total}}{\cos\theta}\right)} \right) \tag{2}$$

$$LAI_{shaded} = LAI_{total} - LAI_{sunlit} \tag{3}$$

Where $\theta$ is the solar zenith angle, $LAI_{total}$ is the total leaf area index of the canopy, and $\Omega$ is the clumping index.

GPP values of sunlit and shaded leaves are calculated using the Farquhar's model (Farquhar et al., 1980) with consideration of the large difference in incident solar irradiance between these two-leaf groups (Chen et al., 2012; Chen et al., 2019). Stomatal conductances of sunlit and shaded leaves are determined separately according to photosynthesis rates of these leaves, atmospheric $CO_2$ concentration, relative humidity and soil moisture (Ball et al., 1987; Ju et al., 2010). The detailed descriptions about the photosynthesis and stomatal conductance modeling approach of BEPS are illustrated in Appendix A1.

The ecosystem COS flux includes both plant COS uptake $F_{COS,plant}$ and soil COS flux exchange $F_{COS,soil}$ (Whelan et al., 2016). In this work, the canopy-level COS plant uptake $F_{COS,plant}$ ($pmol\,m^{-2}\,s^{-1}$) was calculated by upscaling the resistance analog

model of COS uptake (Berry et al., 2013) with the two-leaf upscaling scheme (Chen et al., 1999). Considering the different responses of foliage to diffuse and direct solar radiation (Gu et al., 2002), $F_{COS,plant}$ is calculated as:

$$F_{COS,plant} = F_{COS,sunlit} LAI_{sunlit} + F_{COS,shaded} LAI_{shaded} \tag{4}$$

where $F_{COS,sunlit}$ and $F_{COS,shaded}$ denote the leaf-level COS uptake rate (pmol m$^{-2}$ s$^{-1}$) for sunlit and shaded leaves. The leaf-level COS uptake rate $F_{COS,leaf}$ is determined by the formula (Berry et al., 2013):

$$F_{COS,leaf} = COS_a \left( \frac{1.94}{g_{sw}} + \frac{1.56}{g_{bw}} + \frac{1}{g_{COS}} \right)^{-1} \tag{5}$$

where $COS_a$ represents the COS mole fraction in the bulk air. $g_{sw}$ and $g_{bw}$ are the stomatal conductance and leaf laminar boundary layer conductance to water vapor (H$_2$O). The factors 1.94 and 1.56 account for the smaller diffusivity of COS with respect to H$_2$O. $g_{COS}$ indicates the apparent conductance for COS uptake from the intercellular airspaces, which combined the mesophyll conductance (Evans et al., 1994) and the biochemical reaction rate of COS and carbonic anhydrase (Badger and Price, 1994). It can be calculated as :

$$g_{COS} = \alpha\, V_{cmax} \tag{6}$$

Where $\alpha$ is a parameter that is calibrated to observations of simultaneous measurements of COS and CO$_2$ uptake (Stimler et al., 2012). $V_{cmax}$ is the maximum carboxylation rate of Rubisco. Analysis of these measurements yield estimates of α of ~1400 for C3 and ~7500 for C4 species. With reference the COS modelling scheme of the Simple biosphere model (version 4.2) (Haynes et al., 2020), $g_{COS}$ can be calculated as

$$g_{COS} = 1.4 * 10^3 * (1.0 + 5.33 * F_{C4}) * 10^{-6} F_{APAR}\, f_w V_{cmax} \tag{7}$$

where $F_{C4}$ denotes the C4 plant flag, taking the value of 1 for C4 plants and 0 otherwise. $f_w$ is a soil moisture stress factor describing the sensitivity of $g_{sw}$ to soil water availability (Ju et al., 2006). $F_{APAR}$ is the scaling factor for leaf radiation (Smith et al., 2008), calculated as:

$$F_{APAR} = 1 - e^{(-0.45\, LAI)} \tag{8}$$

The soil COS fluxes are simulated by considering the abiotic and biotic components separately, as by Whelan et al. (2016). We took the soil COS modeling scheme including the parameterizations from Whelan et al. (2016) and Whelan et al. (2022) in this study (see Appendix A2) given that our focus is the COS and GPP relationships and the previous studies have verified this approach over multiple sites with measurements.

## 2.2 Site description

The model was evaluated on seven sites distributed on the Eurasian and American continents in boreal, temperate and subtropical regions based on field observations collected from several studies. Those sites were representative of different climate regions and land cover types (in the model represented by plant function types, and soil textures, as depicted in Table 1).

**Table 1**. Site characteristics. Site identification includes the country initials and a three-letter name for each site. Locations of the sites are provided by the latitude (Lat) and longitude (Lon). PFT stands for plant functional type. ENF and DBF denote evergreen needleleaf forest and deciduous broadleaf forest respectively.

| Site name | Lat (°N) | Lon (°E) | PFT | Soil texture | Year | References |
|-----------|----------|----------|-----|--------------|------|------------|
| AT-Neu | 47.12 | 11.32 | C3 grass | Sandy loam | 2015 | Spielmann et al. (2019) |
| DK-Sor | 55.49 | 11.64 | DBF | Sandy loam | 2016 | Spielmann et al. (2019) |
| ES-Lma | 39.94 | -5.77 | C3 grass | Sandy loam | 2016 | Spielmann et al. (2019) |
| FI-Hyy | 61.85 | 24.29 | ENF | Sandy loam | 2013-2017 | Vesala et al. (2022), Sun et al. (2018) |
| IT-Soy | 45.87 | 13.08 | C3 crop | Silt clay | 2017 | Spielmann et al. (2019), Abadie et al. (2022) |
| US-Ha1 | 42.54 | -72.17 | DBF | Sandy loam | 2012-2013 | Wehr et al. (2017), Commane et al. (2015) |
| US-Wrc | 45.82 | -121.95 | ENF | Loam | 2014 | Rastogi et al. (2018), Shaw et al. (2004) |

## 2.3 Data

Data used in this study include LAI, land cover type, meteorological and soil data, as well as $CO_2$ and COS mole fraction data. The $CO_2$ and COS mole fractions in the bulk air were assumed to be spatially invariant over the globe but to vary annually. The $CO_2$ mole fraction data in this study are taken from the Global Monitoring Laboratory (https://gml.noaa.gov/ccgg/trends/global.html). For the COS mole fraction, we utilized the average of observations from sites SPO (South Pole) and MLO (Mauna Loa, United States) to drive the model. These data are publicly available online at: https://gml.noaa.gov/hats/gases/OCS.html.

### 2.3.1 LAI dataset

The LAI dataset used here are the GLOBMAP global leaf area index product (Version 3) (see GLOBMAP global Leaf Area Index since 1981 | Zenodo) and the Global Land Surface Satellite (GLASS) LAI product (Version 3) (acquired from ftp://ftp.glcf.umd.edu/). They represent LAI at a spatial resolution of 8 km (Liu et al., 2012) and 1 km (Xiao et al., 2016) respectively, and a temporal resolution of 8-day. With reference to the observed LAI at these sites (Wehr et al., 2017; Rastogi et al., 2018; Spielmann et al., 2019; Kohonen et al., 2022), we used GLOBMAP products to drive the BEPS model at most sites (5/7) due to its good agreement with the observed LAI. Specifically, as the GLOBMAP product had considerably underestimated LAI at DK-Sor and was not consistent with the vegetation phenology at ES-Lma during the measurement campaign (Spielmann et al., 2019), GLASS LAI was used at these two sites. In addition, these LAI products were interpolated into daily values by the nearest neighbor method for the simulation.

### 2.3.2 Meteorological dataset

Meteorological data required to force the BEPS model include air temperature, shortwave radiation, precipitation, relative humidity and wind speed. As the simulations were conducted at the site scale, we utilized the FLUXNET2015 data (see https://fluxnet.org for AT-Neu, DK-Sor and ES-Lma, FI-Hyy and US-Ha1) and the AmeriFlux data (see

 for US-Ha1 and US-Wrc). As FLUXNET2015 meteorological data for AT-Neu were only accessible for the period 2002-2012, we conducted a linear fit between its ERA5 (European Centre for Medium-Range Weather Forecasts (ECMWF) Reanalysis v5) data (

) and FLUXNET2015 meteorological data for the corresponding period. Then, we used the fitted parameters to adjust the ERA5 data for 2015, thereby obtaining downscaling information for the meteorological data. In addition, we utilized the FLUXNET data in 2012, and Ameriflux data and ERA5 shortwave radiation data in 2013 to drive the BEPS model at US-Ha1, due to the absence of FLUXNET data in 2013 and the lack of shortwave radiation data of Ameriflux.

### 2.3.3 COS and GPP datasets

The hourly ecosystem COS flux observations were utilized to perform optimization and to evaluate the optimization results. They were derived from existing studies with pre-processing with regard to the data quality check, as listed in Table 1. Specifically, following the recommendations regarding the standardized processing of eddy covariance flux measurements of COS by Kohonen et al. (2020), both the measured and gap-filled COS flux observations are provided in Vesala et al. (2022), and the latter were utilized in this study. To assess the model performance of GPP, the GPP observations were also collected

from FLUXNET (DK-Sor, ES-Lma, FI-Hyy and US-Ha1 in 2012), AmeriFlux (US-Ha1 in 2013), and existing studies (Spielmann et al. (2019) for AT-Neu and IT-Soy and Rastogi et al. (2018) for US-Wrc). Given that only $CO_2$ turbulent flux (FC) or net ecosystem exchange (NEE) data were available at AT-Neu, IT-Soy and US-Ha1 in 2013, a night flux partitioning model (Reichstein et al., 2005) was employed to derive GPP. This model assumes that nighttime NEE represents ecosystem respiration $R_{eco}$, and thus partitions FC or NEE into GPP and $R_{eco}$ based on the semi-empirical models of respiration, which

use air temperature as a driver (Lloyd and Taylor, 1994; Lasslop et al., 2012).

### 2.4 The Monte Carlo-based parameter optimization approach

To evaluate the sensitivity, equifinality and interaction of model parameters, the Monte Carlo-based parameter optimization approach was employed here (Figure 1). The methodology calls for rejecting the concept of a unique global optimum parameter set within some particular model structure, instead recognizing the "equifinality" of parameter sets that exhibit similarly good

performance in producing the observed data (Beven and Freer, 2001). In a Monte Carlo simulation framework, a large number of random sets of parameters are derived across specified parameter ranges (Staudt et al., 2010) and employed to drive the model. Subsequently, model realizations are grouped into behavioral and non-behavioral model runs and associated parameter sets based on the values of the single or multiple performance measures and the predefined threshold value (Houska et al., 2014). The former describes acceptable model realizations conditioned on the available observational data (Blasone et al.,

2008; Beven and Binley, 2014). The latter describes parameter sets that produce behavior inconsistent with observed behavior. Given the gradual transition of performance measures between behavioral and non-behavioral model runs within the Monte Carlo framework, the threshold value used to distinguish between behavioral and non-behavioral parameter sets was often determined by an acceptable samples rate, i.e., ranking model runs and taking the top X% as behavioral (Beven and Binley,

2014). In the past few decades, this approach has been extensively used in ecosystem modeling with multiple parameters to be

calibrated and shown high ability in constraining multiple ecosystem processes (Tonkin and Doherty, 2009; Houska et al., 2014; He et al., 2016; Wu et al., 2019; Wu et al., 2020; Xing et al., 2023).

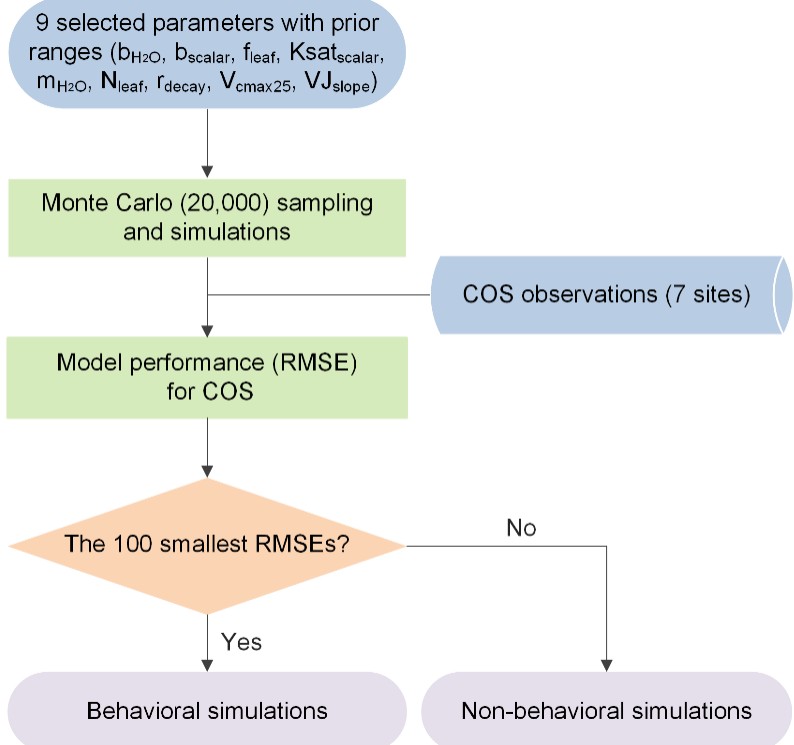

**Figure 1**. Flowchart of the Monte Carlo-based parameter optimization approach used in this study.

### 2.4.1 Parameter selection and sampling strategy

Based on current understanding of COS exchange (Wohlfahrt et al., 2012; Berry et al., 2013; Whelan et al., 2016; Whelan et al., 2018; Cho et al., 2023) and photosynthesis processes (Ball et al., 1987; Raines, 2003; Blankenship, 2021) and related parameter sensitivity studies (Liu et al., 2011; Chen et al., 2012; Chen et al., 2023; Xing et al., 2023; Abadie et al., 2023; Zhu et al., 2023), 9 parameters were selected to be calibrated in this study (for details see Table 2). These parameters are related to formulas describing four processes: 1) photosynthesis ($V_{cmax25}$, $VJ_{slope}$, $N_{leaf}$), 2) soil hydrology ($Ksat_{scalar}$, $b_{scalar}$, $r_{decay}$),

3) stomatal gas exchange ($b_{H_2O}$, $m_{H_2O}$), and 4) energy balance ($f_{leaf}$). Specifically, $Ksat_{scalar}$ and $b_{scalar}$ are scaling factors designed to optimize the saturated hydraulic conductivity (Ksat) and the Campbell parameter (b) for each soil layer in the BEPS model. The prior values and prior ranges for these parameters (Table 2) were chosen based on literature (Jackson et al., 1996; Medlyn et al., 1999; Kattge et al., 2009; Miner et al., 2017; Ryu et al., 2018) and default model settings. Uniform distributions were assigned to all parameters, and 20,000 sets of parameters were generated through random sampling.

**Table 2.** Descriptions of the 9 parameters that were selected to be calibrated. The prior values and prior ranges (in parentheses) of these parameters are given for each plant function type (PFT) or for each soil texture or globally according to the parameter dependency. ENF and DBF denote evergreen needleleaf forest and deciduous broadleaf forest respectively.

| Parameter | Description | Dependent | Prior value and prior range | | | |
|---|---|---|---|---|---|---|
| | | | ENF/ Sand loam | DBF/Slit loam | C3 grass/ Loam | C3 crop |
| $b_{H_2O}$ | The intercept of the Ball-Berry model (mol m$^{-2}$s$^{-1}$) | PFT | 0.0175 (0-1) | 0.0175 (0 -1) | 0.0175 (0 -1) | 0.0175 (0-1) |
| $b_{scalar}$ | The scaling factor of Campbell parameter b (unitless) | Texture | 1 (0.25-1.75) | 1 (0.25-1.75) | 1 (0.25-1.75) | 1 (0.25-1.75) |
| $f_{leaf}$ | Ratio of photosynthetically active radiation to shortwave radiation (unitless) | Global | 0.466 (0.42-0.51) | | | |
| $Ksat_{scalar}$ | The scaling factor of saturated hydraulic conductivity Ksat (unitless) | Texture | 1 (0.25-1.75) | 1 (0.25-1.75) | 1 (0.25-1.75) | 1 (0.25-1.75) |
| $m_{H_2O}$ | The slope of the Ball-Berry model (unitless) | PFT | 8 (2-14) | 8 (2-14) | 8 (2-14) | 8 (2-14) |
| $N_{leaf}$ | Leaf nitrogen content (m$^2$ g$^{-1}$) | PFT | 3.10 (0.40-5.80) | 1.74 (0.32-3.16) | 1.75 (0.23-3.27) | 1.62 (0.40-2.84) |
| $r_{decay}$ | Decay rate of root distribution (unitless) | PFT | 0.95 (0.80-0.99) | 0.97 (0.80-0.99) | 0.96 (0.80-0.99) | 0.95 (0.80-0.99) |
| $V_{cmax25}$ | Maximum carboxylation rate of Rubisco at 25 ℃ (μmol m$^{-2}$s$^{-1}$) | PFT | 62.5 (13.1-111.9) | 57.7 (15.3-100.1) | 78.2 (16-140.4) | 100.7 (27.5-173.9) |
| $VJ_{slope}$ | Slope of the $V_{cmax}$ and $J_{max}$ (maximum electron transport rate) relationship (unitless) | PFT | 2.39 (1-4) | 2.39 (1-4) | 2.39 (1-4) | 2.39 (1-4) |

## 2.4.2 Selection of behavioral simulations

To measure the agreement between model simulations and observations, a variety of performance metrics have been proposed and utilized in previous studies (Beven and Binley, 1992; Moradkhani et al., 2005; Staudt et al., 2010). In this study, we employed the root mean square error (RMSE) to distinguish between behavioral and non-behavioral simulations.

$$RMSE = \sqrt{\frac{1}{N}\sum_{i=1}^{N}(obs_i - sim_i)^2} \qquad (3)$$

where N is the total number of observations."obs" and "sim" denote the observations and simulations, respectively. $sim_i$ denotes the simulation corresponding to the $i$ th observation $obs_i$.

Specifically, here we chose an acceptable samples rate of 0.5%, i.e., the top 100 model runs with the lowest RMSE values for COS as behavioral simulations. Thus, the deterministic model prediction is given by the ensemble mean of the 100 behavioral simulations.

## 2.5 Uncertainty quantification

The model prediction limits or uncertainty bounds can be determined by forming the cumulative density function (CDF) of the ensemble of simulations (Beven and Binley, 2014), normally chosen at the 5 % and 95 % confidence level in most of the previous studies (Blasone et al., 2008). Similarly, we chose the 5 % and 95 % quantiles of the 20,000 simulations and the 100 behavioral simulations to quantify the model output uncertainty in this study.

## 2.6 Parameter sensitivity

In order to take full advantage of the Monte Carlo simulations, a density-based global sensitivity analysis approach (Plischke et al., 2013) was used to investigate the sensitivity of COS and GPP simulations to the selected model parameters via the Sensitivity Analysis Library (SALib) (Iwanaga et al., 2022). This approach aims at assessing the influence of the entire input distribution on the entire output distribution without reference to a particular moment of the output (Borgonovo, 2007). According to Borgonovo (2007), the sensitivity index ($\delta$) is always between 0 and 1, it equals 0 if the output is not dependent upon the model parameter, and it equals unity if all model parameters are considered.

## 2.7 Parameter uncertainty

Due to the functional and structural complexity of ecosystems, ecosystem models often require a substantial number of parameters to realize the modeling of various ecosystem processes, and some parameters are compensating each other (Mo et al., 2008). While the parameter interactions related to photosynthesis have been systematically studied (Tang and Zhuang, 2009; Lu et al., 2013; Wu et al., 2019; Xing et al., 2023), the parameter interactions related to COS flux simulation have not been reported. Based on the Monte Carlo-based methodology, the numerous behavioral parameter sets around the "optimum" (Beven and Freer, 2001) provide us with the opportunity to analyze the interactions between the selected parameters. In this study, the Pearson correlation coefficient and the confidence level were employed to identify the parameter interactions. Parameter identifiability (PI) is the concept of whether uncertain parameters can be correctly estimated from the observed data (Yi et al., 2019). The failure in PI is supposed to be caused by 'over-parameterization' and parameter interactions (due to high nonlinearity of model equations) (Gan et al., 2014). Inspired by Yi et al. (2019) who used likelihood confidence interval as a measure of PI, here we used parameter distribution range for the same purpose. Taking into account the influence of the prior distribution of the behavioral parameter sets, the PI is defined as the reduction of the parameter range width. Hence, a large value of PI indicates that the parameter is well identified in the optimization process.

# 3 Results

## 3.1 Parameter sensitivity

The sensitivity indexes of COS and GPP simulations to the model parameters for the seven sites are illustrated in Fig. 1. It can be seen that both COS and GPP simulations exhibit high sensitivity to leaf nitrogen content ($N_{leaf}$) and the maximum carboxylation rate of Rubisco at 25 °C ($V_{cmax25}$), while showing low sensitivity to energy balance related parameter $f_{leaf}$ as well as soil hydrology related parameters (including $b_{scalar}$, $Ksat_{scalar}$ and $r_{decay}$). With the average values of sensitivity index of 0.11 and 0.10, the photosynthesis related parameter $VJ_{slope}$ as well as stomata conductance related parameter $m_{H_2O}$ can significantly impact the simulation of GPP. However, those parameters do not exhibit high sensitivity in the modeling of COS. Our results also highlight the crucial role of the intercept of the Ball-Berry model ($b_{H_2O}$) in the modeling of COS, yet its impact on the simulation of GPP is limited. In summary, our results suggest that the simulated COS and GPP share some similarities in their sensitivities to parameters, but there are also notable differences. Specifically, the parameters $m_{H_2O}$ and $VJ_{slope}$ strongly influence GPP simulations but have minimal impacts on COS simulations. Conversely, the parameter $b_{H_2O}$ plays a more crucial role in COS simulation.

With mean values of 0.33, 0.29 and 0.09 respectively, the sensitivity indexes of COS simulations to $N_{leaf}$, $V_{cmax25}$ and $b_{H_2O}$ are much larger than those of GPP simulations. However, the patterns of the sensitivity of these parameters for COS and GPP simulations are very similar across these sites. Our results reveal that the simulated COS and GPP are more sensitive to $N_{leaf}$, while less influenced by $V_{cmax25}$. In comparison to other sites, the role of $V_{cmax25}$ in the simulation of COS and GPP at IT-Soy is less significant. Additionally, we observed that $m_{H_2O}$ holds greater importance in the modeling of GPP at US-Wrc than at other sites. Moreover, our results suggest that the modeling of GPP at deciduous broadleaf forest sites (DK-Sor and US-Ha1) are more sensitive to $VJ_{slope}$ while less sensitive to $m_{H_2O}$ than at other sites.

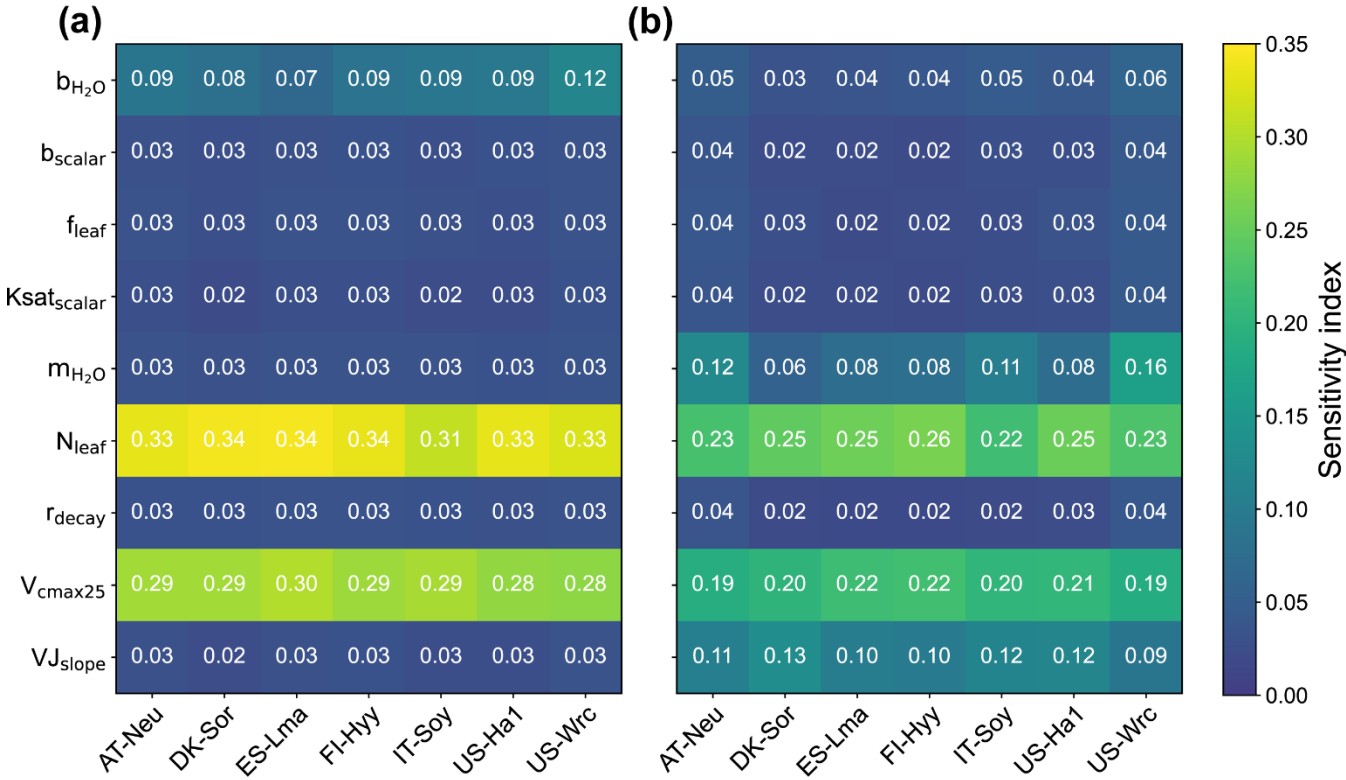

**Figure 1.** Sensitivity indexes of the modelled ecosystem COS fluxes (**a**) and GPP (**b**) to model parameters.

## 3.2 Posterior parameter distributions

The cumulative frequency distribution as well as the boxplots of each of the parameters for the 0.5 % best runs were plotted in Fig. 2, with a comparison to the uniform parameter distributions and the prior parameter values. As shown in Fig. 2, the posterior distributions of these parameters differ significantly, indicating that the response of these parameters to the assimilation of COS is quite different. Our results demonstrated that COS fluxes have similar constraining effects to the same parameters in different ecosystems although the posterior distributions of the same parameter at different sites depicted variations. In general, parameters related to plant growth and stomatal conductance were strongly constrained by the assimilation of COS, while the parameters related to energy balance as well as soil hydrology were inadequately constrained. With distinct shape and remarkably narrow range of the cumulative frequency curves, $b_{H_2O}$ (the intercept of the Ball-Berry model, representing minimum stomatal conductance) was strongly constrained by the assimilation of COS in this study. For most sites (AT-Neu, DK-Sor, FI-Hyy, IT-Soy and US-Ha1), the values of $b_{H_2O}$ were confined to a very limited range of 0 to 0.08 mol m$^{-2}$ s$^{-1}$. At these five sites, the average values of the posterior $b_{H_2O}$ all located from 0.01 to 0.03 mol m$^{-2}$ s$^{-1}$, aligning well with the default value of $b_{H_2O}$ (0.0175 mol m$^{-2}$ s$^{-1}$) for the BEPS model. By contrast, with posterior $b_{H_2O}$ values ranging from 0.03 to 0.18 and 0.03 to 0.91 mol m$^{-2}$ s$^{-1}$, the default value of $b_{H_2O}$ for the BEPS model was rejected by

the assimilation of COS at ES-Lma and US-Wrc. Despite the broad distribution of posterior $b_{H_2O}$ at US-Wrc, the cumulative frequency curve still indicates that $b_{H_2O}$ is well-constrained at this site, with 80 % of the posterior $b_{H_2O}$ located in a narrow range of 0.15 to 0.50 mol m$^{-2}$ s$^{-1}$. Overall, our results are reasonable as literature-documented values of $b_{H_2O}$ are highly variable and they align well with the compilation provided by Miner et al. (2017), in which more than 5/6 of the $b_{H_2O}$ values are located between 0 and 0.18 mol m$^{-2}$ s$^{-1}$, and about half are located between 0 and 0.04 mol m$^{-2}$ s$^{-1}$. Moreover, the mean values of posterior $b_{H_2O}$ for most (5/7) sites are larger than the default $b_{H_2O}$ value of the BEPS model, suggesting that the current $b_{H_2O}$ value utilized in BEPS may be underestimated.

Identified as the most sensitive parameters in COS modeling, the plant growth related parameter $V_{cmax25}$ and $N_{leaf}$ were generally well constrained in this study. However, unlike $b_{H_2O}$, which is strongly constrained at all sites, the posterior cumulative frequency curves of $V_{cmax25}$ and $N_{leaf}$ exhibit considerable variation across sites. Except for US-Ha1 and US-Wrc, the posterior $V_{cmax25}$ and $N_{leaf}$ were mostly distributed in the upper half of the parameter range. Particularly, all of the lower half values of $V_{cmax25}$ and $N_{leaf}$ were excluded by the behavioral parameter sets at ES-Lma. In contrast, the posterior cumulative frequency curves of $V_{cmax25}$ deviated slightly from the original uniform distribution at US-Ha1, indicating that they are not well-constrained by the assimilation of COS. As for US-Wrc, both the largest 7 % and smallest 4 % values of $N_{leaf}$ are effectively excluded by the assimilation of COS.

Another stomatal conductance-related parameter, $m_{H_2O}$, demonstrated effective constraint through COS assimilation at specific sites (AT-Neu, DK-Sor, ES-Lma, FI-Hyy, and IT-Soy), with parameter range width reductions comparable to $V_{cmax25}$ and $N_{leaf}$. However, at US-Ha1 and US-Wrc, the posterior cumulative frequency curves of $m_{H_2O}$ show minimal deviation from the original uniform distribution. Nevertheless, the optimization of $m_{H_2O}$ is generally achievable through COS assimilation, as supported by our results in good agreement with the compilation of Miner et al. (2017), in which the average historical values of $m_{H_2O}$ grouped by PFT (referred to as the PFT-grouping values below) are provided. As indicated in Table 3, the average absolute bias between the default $m_{H_2O}$ and the PFT-grouping value reached as high as 2.87 for these sites. Through COS assimilation, the mean absolute bias was reduced to 2.59.

**Table 3.** Mean posterior $m_{H_2O}$ values for seven study sites in comparison with the default values and the PFT-grouping values (mean ± standard deviation) in Miner et al. (2017). Within the compilation of Miner et al. (2017), FI-Hyy and US-Wrc are classified under the PFT of evergreen gymnosperm tree, while DK-Sor and US-Ha1 fall under the PFT of deciduous angiosperm tree.

| Site name | AT-Neu | DK-Sor | ES-Lma | FI-Hyy | IT-Soy | US-Ha1 | US-Wrc |
|---|---|---|---|---|---|---|---|
| Default | 8 | 8 | 8 | 8 | 8 | 8 | 8 |
| This study | 6.41 | 9.53 | 10.37 | 5.13 | 9.33 | 8.00 | 7.76 |
| Miner 2017 | 13.3 ± 3.1 | 8.7 ± 5.1 | 13.3 ± 3.1 | 6.7 ± 2.5 | 13.5 ± 3.1 | 8.7 ± 5.1 | 6.7 ± 2.5 |

The photosynthesis-related parameters $VJ_{slope}$ and $f_{leaf}$ also influence COS simulation. However, the posterior distributions of $f_{leaf}$ resemble the original uniform distribution, suggesting that it is not a crucial parameter for COS simulations. The

posterior cumulative frequency curve of $VJ_{slope}$ also generally deviates slightly from the uniform distribution. Yet, at DK-Sor and US-Ha1, more than two-thirds of the posterior $VJ_{slope}$ values are situated in the upper half of the parameter range, indicating that $VJ_{slope}$ can also be well-constrained by the assimilation of COS in specific cases.

Among these seven sites, the soil hydrology-related parameters $Ksat_{scalar}$ and $b_{scalar}$ did not exhibit a strong response during the assimilation of COS. However, the posterior cumulative frequency curves of $r_{decay}$ show notable deviations from the uniform distribution in certain cases. At US-Wrc, higher values of $r_{decay}$ are more prevalent within the behavioral parameter sets, leading to the posterior mean of $r_{decay}$ much greater than the prior mean. Moreover, the largest 14 % values of $r_{decay}$ are effectively excluded by the assimilation of COS at IT-Soy.

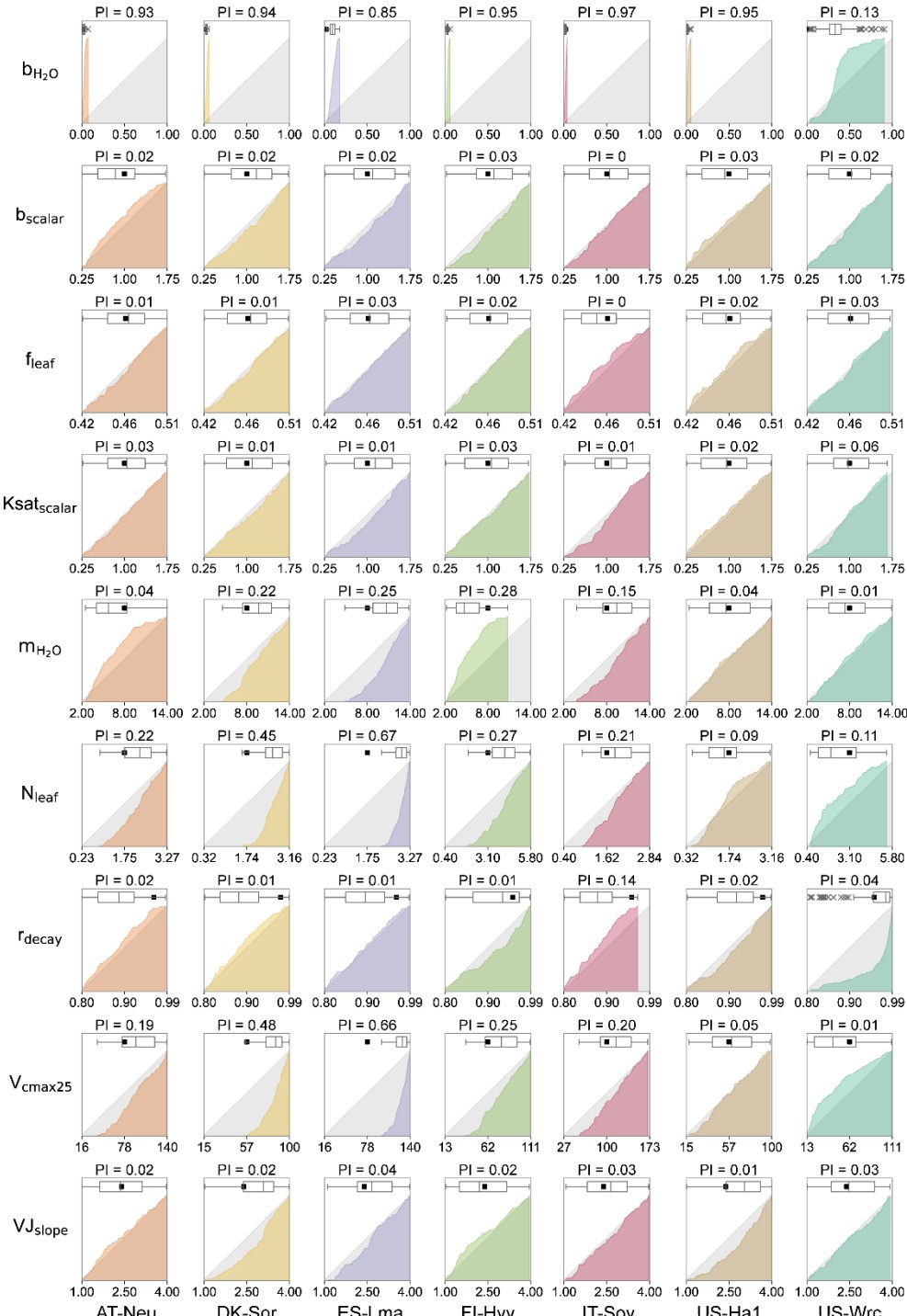

**Figure 3.** Cumulative frequency distributions and boxplots for the posterior model parameters obtained by COS assimilation. The grey area represents uniform parameter distributions, while the colored areas denote posterior CDF distributions, with parameters for different sites

represented using different colors. The box extends from the first quartile to the third quartile of the parameter values, with a line at the median. "×" markers denote outliers, and the whiskers represent the lowest or highest parameter values excluding any outliers. The black squares represent the prior parameter values, and the axis ranges denote the prior ranges of the parameters. PI denotes parameter identifiability, defined as the reduction of the parameter range width.

## 3.3 The optimization performance in COS fluxes

The posterior simulated COS fluxes were evaluated against the prior simulations and observations. Table 4 lists the mean RMSEs and range widths of the prior and posterior simulated COS fluxes for all the sites. The $RMSE_{mean}$ of the posterior COS simulations are smaller than the prior, and the mean RMSE reduction for all sites is 35.51 % ± 13.72 % (mean ± SD). At the same time, the simulation range widths of COS fluxes are also well constrained, with a mean reduction of 81.46 % ± 7.32 % from the prior. The reduction of $RMSE_{mean}$ and range width is particularly significant in US-Wrc, with a value of 61.14 % and 94.70 %, respectively.

In Fig. 4, the daily and monthly variations of COS during the observation period at each site are shown. It can be observed that both the prior and posterior simulations are able to accurately capture the daily variation or the seasonal cycle of COS across these sites, with the exception of IT-Soy. As IT-Soy is a temporary observatory with no continuous in-situ meteorological observations available, the ERA-5 meteorological data were used to drive the model for this site, resulting in the simulation not being able to characterize the COS changes very well. Although the simulations perform well in modeling the variations of COS for other sites, our results also suggest that they tend to underestimate the magnitude of COS fluxes at both ends of the growing season (e.g. Fig. 4d). Furthermore, the model markedly underestimates the magnitude of COS during rainy days (DOY 126-134) at ES-Lma (Fig. 4c). These findings suggest substantial deficiencies in modeling the mechanistic process of COS exchange. Nevertheless, it can be stated that the fusion of COS observations with the BEPS model has the capacity in constraining the predictive uncertainty of COS, as evidenced by significantly reduced uncertainty bounds that largely encapsulate observations.

The prior simulations significantly underestimate the COS fluxes at ES-Lma, with the ensemble mean of prior simulations being only about one-third of that of the observations. After optimization, the simulated COS fluxes show a substantial increase and generally align with the observations. However, some observed peaks are still not included in the posterior simulation uncertainty bounds. In contrast, the prior simulations tend to overestimate COS fluxes at forest site FI-Hyy, US-Ha1 and US-Wrc. At US-Wrc and US-Ha1, the ensemble means of prior simulations are 65.70 % and 64.81 % larger than the observations. The assimilation of COS effectively corrected the overestimation but, at the same time, led to a slight underestimation of the simulated COS for US-Wrc. With the down-regulation of COS simulations, the model-observation difference at both ends of the growing season for FI-Hyy further increased. Particularly, significant underestimation is found in the posterior simulations in 2017 for FI-Hyy, despite the posterior simulations showing a remarkable improvement in reproducing COS fluxes over the entire period (2013-2017). As the prior simulations neither noticeably overestimate nor underestimate, there is little difference between the ensemble mean of the prior and posterior simulations at the remaining three sites (AT-Neu, DK-Sor and IT-Soy).

Nevertheless, the assimilation of COS resulted in a remarkable reduction in both $RMSE_{mean}$ and uncertainty bounds for COS simulations at these sites, with mean reductions of 23.93 % and 75.11 %, respectively.

Overall, there are considerable uncertainties in the prior simulations, with the uncertainty bounds comparable to or much larger than the uncertainties of observations, and the ensemble mean strongly deviates from observations in some sites, i.e., ES-Lma. Our results suggest that significant improvement in both the ensemble mean and predictive uncertainty of COS simulations can be achieved through the addition of the information provided by the COS observations with the Monte Carlo-based parameter optimization approach , especially for evergreen needleleaf forest sites. However, limited by various factors, such as uncertainty in model-driven data and model structure (Cho et al., 2023), currently the model often underestimates the simulation at both ends of the growing season, and lacks proficiency in modeling the magnitude of COS during rainy days.

**Table 4.** Comparison of model performance indices for the prior and posterior COS simulations. The $RMSE_{mean}$ of the prior and posterior simulations are the mean values of the RMSEs of 20,000 prior COS simulations and 100 behavioral COS simulations with COS observations, respectively. The range widths of the prior and posterior COS simulations are defined as the mean values of the difference between the 95th and 5th percentile of the prior and posterior simulations, respectively. The reduction (%) of $RMSE_{mean}$ and range width is calculated as (1-posterior/prior) × 100.

| Site name | $RMSE_{mean}$ (pmol m$^{-2}$ s$^{-1}$) | | | Range width (pmol m$^{-2}$ s$^{-1}$) | | |
|---|---|---|---|---|---|---|
| | Prior | Posterior | Reduction (%) | Prior | Posterior | Reduction (%) |
| AT-Neu | 24.10 | 14.30 | 40.67 | 46.40 | 6.79 | 85.36 |
| DK-Sor | 32.69 | 24.07 | 26.36 | 45.41 | 13.33 | 70.64 |
| ES-Lma | 17.10 | 14.66 | 14.26 | 10.35 | 2.75 | 73.47 |
| FI-Hyy | 15.87 | 10.87 | 31.52 | 20.96 | 3.88 | 81.50 |
| IT-Soy | 16.49 | 11.35 | 31.16 | 27.26 | 5.12 | 81.21 |
| US-Ha1 | 30.08 | 17.02 | 43.44 | 50.47 | 8.42 | 83.31 |
| US-Wrc | 36.76 | 14.28 | 61.14 | 78.04 | 4.13 | 94.70 |

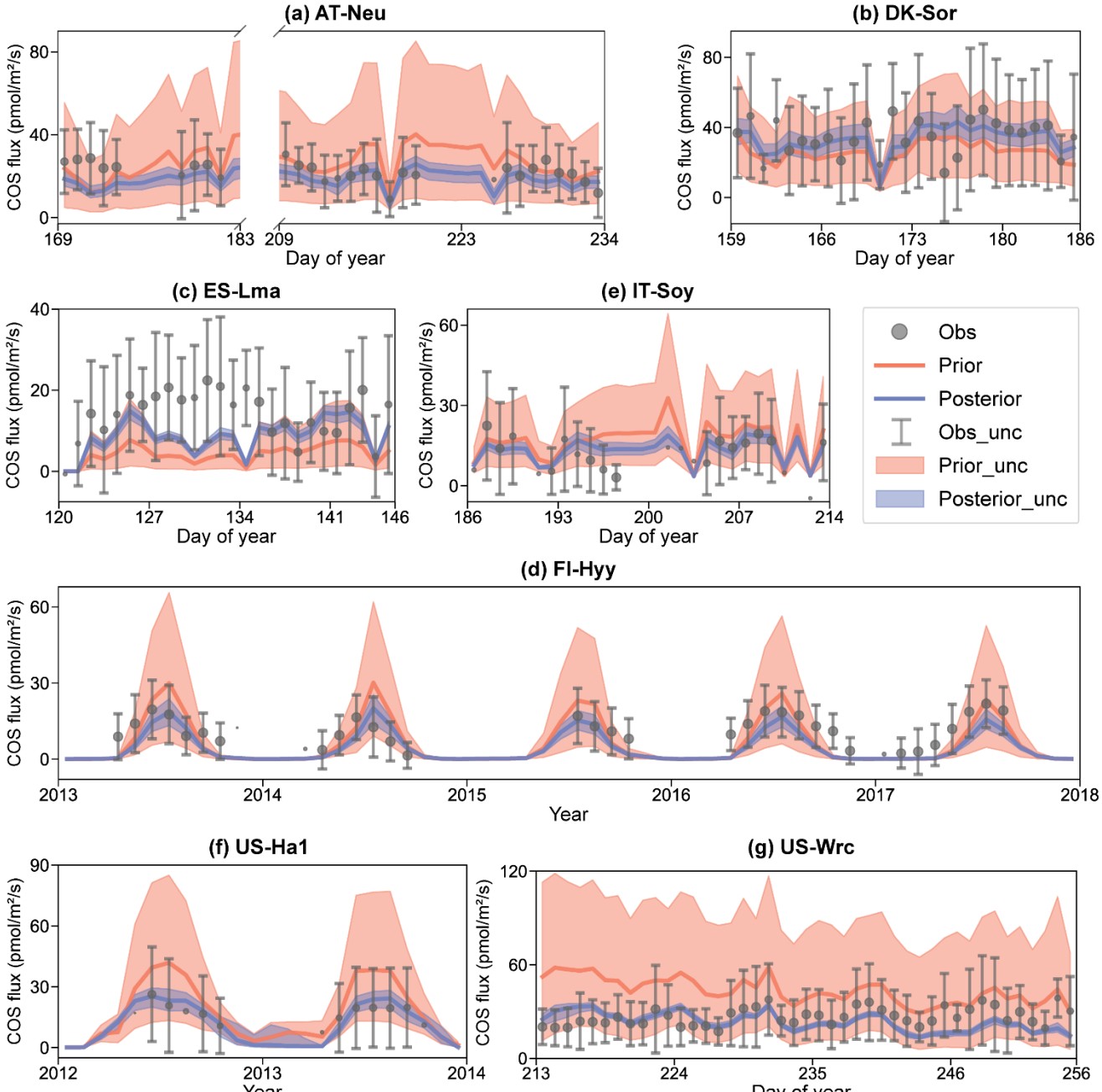

**Figure 4.** Comparison of prior and posterior simulated ecosystem COS fluxes. The ensemble means of the prior (red) and posterior (blue) simulations are plotted around the uncertainty bounds (5th and 95th quantile). The mean observed COS and its uncertainty (estimated by the standard deviation) are represented by black dots with error bars. The means and uncertainties of these hourly observations and simulations are calculated and plotted on a daily or monthly scale. Error bars are not plotted when more than 3/4 of the observations are missing. The subplot numbers are assigned based on the alphabetical order of the site names.

## 3.4 The performance of simulated GPP

The mean RMSEs and range widths of both prior and posterior simulated GPP for all sites are presented in Table 5. With
390 reduction ratios of $RMSE_{mean}$ ranging from 20.16 % to 64.12 %, the assimilation of COS effectively enhanced the model
performance of GPP to varying degrees. Concurrently, the range widths of GPP simulations were well confined, exhibiting a
mean reduction ratio of 65.81 % ± 6.77 %. The maximum reduction in $RMSE_{mean}$ for GPP occurred at US-Wrc, aligning with
the substantial improvement observed in the posterior simulated COS at this site. In contrast, a relatively limited impact on
improving the prediction of GPP was observed at FI-Hyy, as evidenced by both the smaller reduction in $RMSE_{mean}$ and range
width of GPP simulations.

The BEPS model demonstrated excellent performance in capturing the daily variation and seasonal cycle of GPP, as illustrated
in Fig. 5. However, similar to the COS simulations, the ensemble averages of the prior simulated GPP notably deviated from
observations at several sites. For example, at DK-Sor and ES-Lma, the ensemble averages of the prior simulated GPP were
only approximately half of the observations. After the assimilation of COS, GPP simulations exhibited a significant increase,
aligning well with observations at DK-Sor and ES-Lma. Conversely, substantial overestimation in prior GPP simulations was
effectively corrected through the assimilation of COS at US-Wrc, resulting in a remarkable enhanced modeling performance
in both RMSE and range width. For FI-Hyy and US-Ha1, minimal differences were observed between the ensemble mean of
prior and posterior simulations, as the ensemble mean of prior simulated GPP had already consistently fit the observations.
Nevertheless, our results highlight notable enhancements in the predictive uncertainty of GPP through COS assimilation at
405 these two sites. In Fig. 5d, it is evident that, likely due to the absence of in situ meteorological data at IT-Soy, GPP trends are
not well represented, although the ensemble averages of the GPP simulations are very close to the observations in magnitude.
However, with a reduction of range width as high as 74.72 %, our finding suggests that the assimilation of COS can
significantly reduce the predictive uncertainty of GPP, despite the presence of substantial meteorological data uncertainty.

**Table 5.** Comparison of model performance indices for the prior and posterior GPP simulations. The $RMSE_{mean}$ of the prior and posterior
simulations are the mean values of the RMSEs of 20,000 prior GPP simulations and 100 behavioral GPP simulations with GPP observations,
respectively. The range widths of the prior and posterior GPP simulations are defined as the mean values of the difference between the 95th
and 5th percentiles of the prior and posterior simulations, respectively. The reduction (%) of $RMSE_{mean}$ and range width is calculated as
(1-posterior/prior) * 100.

| Site name | $RMSE_{mean}$ (µmol m$^{-2}$ s$^{-1}$) | | | Range width (µmol m$^{-2}$ s$^{-1}$) | | |
|---|---|---|---|---|---|---|
| | Prior | Posterior | Reduction (%) | Prior | Posterior | Reduction (%) |
| AT-Neu | 13.52 | 9.48 | 29.90 | 27.14 | 8.41 | 69.01 |
| DK-Sor | 15.39 | 7.08 | 54.00 | 19.65 | 7.19 | 63.39 |
| ES-Lma | 7.35 | 4.63 | 37.06 | 11.99 | 4.75 | 60.39 |
| FI-Hyy | 5.14 | 4.10 | 20.16 | 8.51 | 3.98 | 53.20 |
| IT-Soy | 10.98 | 7.19 | 34.57 | 20.92 | 5.29 | 74.72 |
| US-Ha1 | 8.05 | 4.50 | 44.14 | 12.09 | 3.51 | 70.97 |

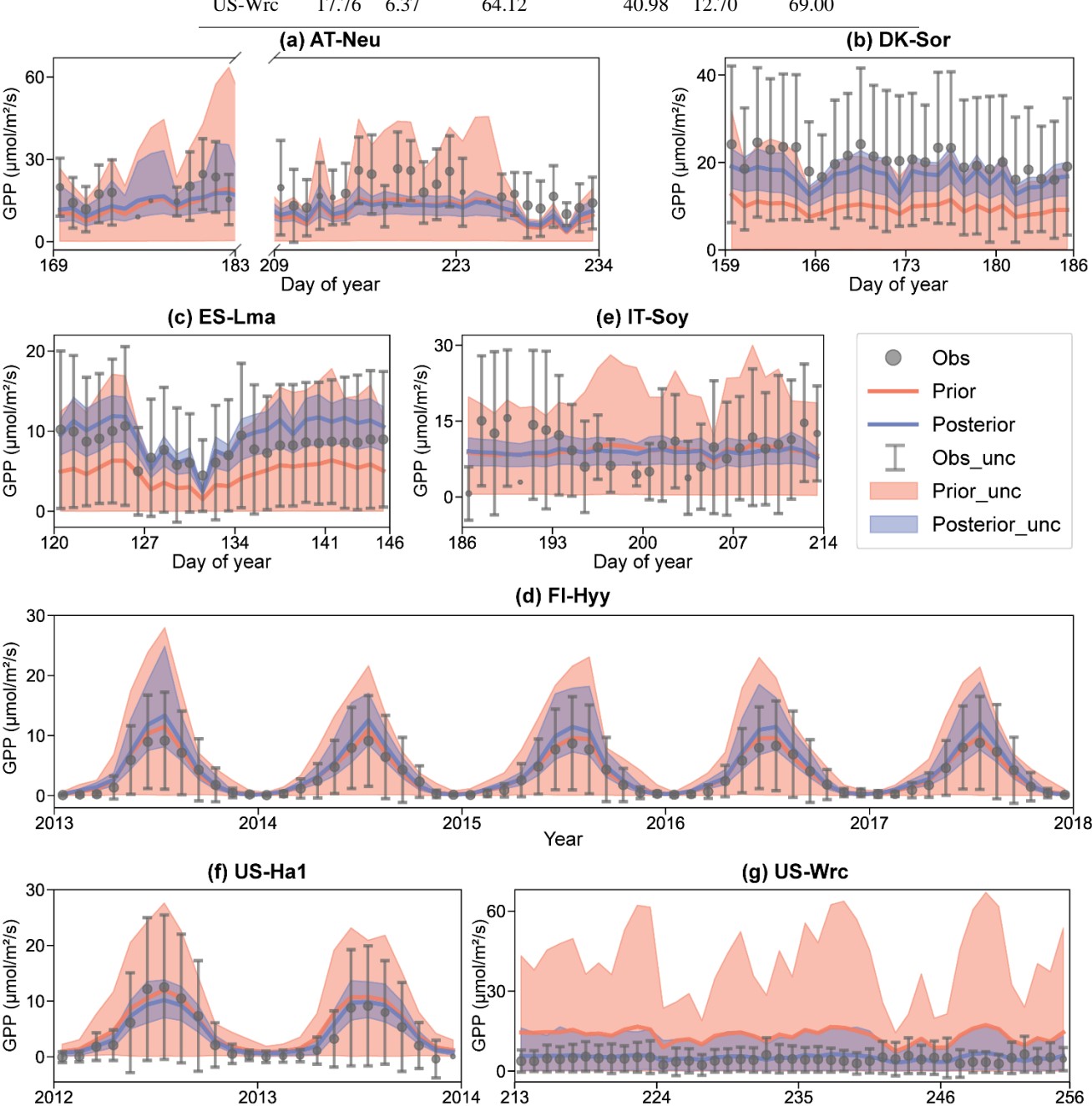

**Figure 5.** Comparison of prior and posterior simulated GPP. The ensemble means of the prior (red) and posterior (blue) simulations are plotted around the uncertainty bounds (5th and 95th quantile). The mean observed GPP and its uncertainty (estimated by the standard deviation) are represented by black dots with error bars. The means and uncertainties of these hourly observations and simulations are calculated and plotted on a daily or monthly scale. Error bars are not plotted when more than 3/4 of the observations are missing. The subplot numbers are assigned based on the alphabetical order of the site names.

## 4 Discussion

### 4.1 Parameter sensitivity

As mentioned before, here we utilize the conductance analog model proposed by Berry et al. (2013) to simulate COS plant uptake. Thus, it is not surprising that both the stomatal conductance related parameter $b_{H_2O}$ and $m_{H_2O}$ would impact the modeling of COS flux. Considering the stress of soil moisture on stomatal conductance, the stomatal conductance was calculated by a modified version (Woodward et al., 1995; Ju et al., 2010) of the Ball-Berry model (Ball et al., 1987) based on the close relationship of stomatal conductance and photosynthesis rate. Consequently, both the soil hydrology related parameters and the photosynthesis related parameters can ultimately play roles in the simulation of COS plant uptake by influencing the modeling of the stomatal conductance.

It has been recognized that the photosynthetic capacity simulated by terrestrial ecosystem models is highly sensitive to $V_{cmax}$, $J_{max}$, and light conditions (Zaehle et al., 2005; Bonan et al., 2011; Rogers, 2014; Sargsyan et al., 2014; Koffi et al., 2015; Rogers et al., 2017; Xing et al., 2023). Our study corroborates these findings, highlighting the pronounced sensitivity of simulated GPP to $V_{cmax25}$, followed by $VJ_{slope}$ and $f_{leaf}$. Moreover, our results reveal that the COS simulations are not notably sensitive to $f_{leaf}$ and $VJ_{slope}$ while $V_{cmax25}$ plays a crucial role in the modeling of COS. It is because $V_{cmax25}$ not only affects the estimation of stomatal conductance through photosynthesis, but also is used to characterize the apparent conductance for COS uptake from the intercellular airspaces, as both mesophyll conductance and carbonic anhydrase activity tend to scale with $V_{cmax}$ (Badger and Price, 1994; Evans et al., 1994; Berry et al., 2013). Yet, as the hydrolysis reaction of COS by carbonic anhydrase is not dependent on light, $VJ_{slope}$ and $f_{leaf}$ do not play any roles in the modeling of apparent conductance and thus have little effect on the simulation of COS.

As the COS plant uptake and photosynthesis are tightly coupled through stomata, one would naturally expect similar sensitivity in simulated COS and GPP to stomatal conductance related parameters $m_{H_2O}$ and $b_{H_2O}$. However, the relationship between COS and stomatal conductance significantly differs from that between GPP and stomatal conductance within the model (e.g., Eq. 5 and the Ball-Berry model). Consequently, a notable difference in sensitivity between simulated GPP and COS to $m_{H_2O}$ and $b_{H_2O}$ was identified in this study. Specifically, $m_{H_2O}$ exhibited more pronounced effects on photosynthesis, while $b_{H_2O}$ played a crucial role in the simulation of COS.

Given that a significant portion of nitrogen is invested in the photosynthetic machinery (Mu and Chen, 2021), there exists a close association between leaf nitrogen content and leaf photosynthetic capacity (Sage and Pearcy, 1987). Additionally, the well-established relationship between leaf nitrogen content and carboxylation capacity (Kattge et al., 2009; Lu et al., 2022) further emphasizes this connection. Specifically, carboxylation capacity in leaf scale is assumed to be linearly related to leaf nitrogen content in the BEPS model (Medlyn et al., 1999; Chen et al., 2012). Consequently, both $V_{cmax25}$ and $N_{leaf}$ play crucial roles in influencing carboxylation capacity, thus having a substantial impact on the simulation of COS.

The soil hydrology related parameters can also affect the simulation of COS plant flux as we take the stress effect of soil moisture on both stomatal conductance and mesophyll conductance into account (Ju et al., 2010; Knauer et al., 2020). These parameters also affect the modeling of COS soil exchange since soil moisture is a significant factor in COS soil biotic flux (Whelan et al., 2016). However, given the smaller magnitude of soil COS exchange compared to plant uptake (Whelan et al., 2018) and the minimal impact of soil moisture stress on photosynthetic capacity (Ma et al., 2022), these soil hydrology relevant parameters do not significantly influence the modeling of COS.

## 4.2 Parameter interactions

For all seven sites, Pearson correlation coefficients and confidence levels between the selected parameters were calculated, as depicted in Fig. 6. Generally, each site exhibits approximately 3 to 8 parameter combinations with significant correlations ($p < 0.05$). A total of 8 parameter combinations demonstrate significant correlations at more than one site, while 11 parameter combinations exhibit significant correlations at only one site. Specifically, with a mean correlation coefficient of $-0.55 \pm 0.14$ (negative value representing a negative correlation), the correlations between $V_{cmax25}$ and $N_{leaf}$ are very significant ($p < 0.01$) at all sites, indicating a robust interaction between them. In addition to $V_{cmax25}$ and $N_{leaf}$, four parameter combinations show highly significant correlations ($p < 0.01$) at a minimum of two sites, they are $b_{H_2O}$ and $m_{H_2O}$, $m_{H_2O}$ and $V_{cmax25}$, $m_{H_2O}$ and $N_{leaf}$, $b_{scalar}$ and $VJ_{slope}$ respectively. Such results indicate the strong interactions among parameters related to stomatal conductance, photosynthesis as well as soil hydrology, even if some of them do not significantly impact the modeling of COS.

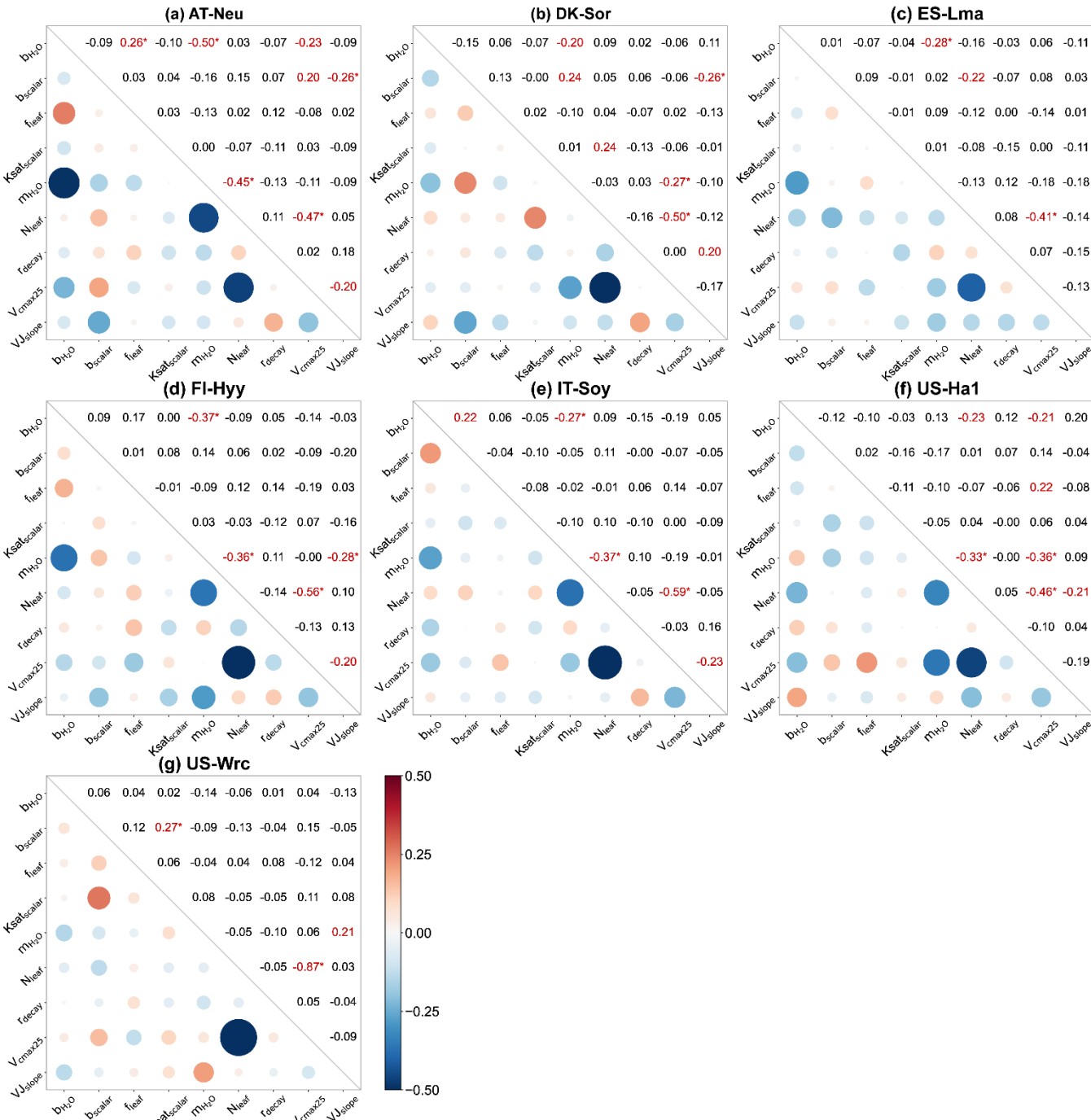

**Figure 6.** Parameter correlation matrix plots with significance levels between the parameters of the behavioral parameter sets. Correlation coefficients are shown in red font when the confidence level is greater than 95 % (p < 0.05), with a superscript "*" indicating the confidence level greater than 99 % (p < 0.01).

We observed substantial variations in parameter interactions across different sites. For instance, at AT-Neu, $m_{H_2O}$ and $N_{leaf}$ exhibited a highly significant negative correlation with a correlation coefficient as high as -0.45. However, these two parameters seemed irrelevant at DK-Sor with a correlation coefficient of only -0.03. As for soil hydrology related parameters, none of them showed significant correlations with any parameter at IT-Soy, FI-Hyy and US-Ha1, yet there were four parameter combinations related to them significantly correlated at DK-Sor (Fig. 6b). Furthermore, while $V_{cmax25}$ is highly correlated with $N_{leaf}$ at all sites, the correlation coefficients varied considerably, ranging from -0.41 to -0.87.

We also observed interactions not only between two parameters but also among several parameters (e.g., $b_{scalar}$ - $V_{cmax25}$ - $VJ_{slope}$ in Fig. 6a). Fig. B1 showcases the intricate interactions among multiple parameters relevant to COS simulations and illustrates the distribution of behavioral parameter sets. The parameter combinations depicted in Fig. B1 are particularly representative as they originate from diverse sites and include nearly all highly significant correlated combinations. Overall, since the six plant growth related parameters used in this study are positively correlated with the simulation of COS, they consistently constrain each other, demonstrating a negative correlation, as shown in Fig. 6a, Fig. 6c, and Fig. 6d. However, due to the nonlinearity of the model, there is not a simple linear relationship between these parameters. For example, at AT-Neu, where the COS observations notably exceed the ensemble mean of prior simulations, $V_{cmax25}$ and $VJ_{slope}$ exhibit a non-linear correlation, but both tend to be distributed near their upper limits (Fig. B1).

### 4.3 Parameter identifiability

As the parameter identifiability is quantified based on the range of the behavioral parameter, its results were presented in Fig. 3 along with the plots of the cumulative likelihood distributions of the behavioral parameters. These results underscore the remarkable ability of COS assimilation to identify $b_{H_2O}$, with a mean PI of $b_{H_2O}$ as high as $0.81 \pm 0.28$. Identified as the most sensitive parameters for COS modeling, $V_{cmax25}$ and $N_{leaf}$ also exhibit remarkable identifiability, with mean PIs of $0.29 \pm 0.19$ and $0.26 \pm 0.22$ respectively. $m_{H_2O}$ demonstrate varying levels of identifiability, with PIs ranging from 0.01 to 0.28. In contrast, the light reaction related parameters $VJ_{slope}$ and $f_{leaf}$ are not well identified, with the maximum value of PI of only 0.04. The soil hydrology related parameters $b_{scalar}$, $Ksat_{scalar}$ and $r_{decay}$ are also generally unidentifiable. Notably, $r_{decay}$ is well identified at IT-Soy, in which its PI value (0.14) is approximately seven times that of the other sites.

In this study, the identifiability of a parameter closely related to the sensitivity of COS simulations to the parameter, although it is known to be influenced by model over-parameterization and parameter interactions (Gan et al., 2014). For instance, at ES-Lma, where the COS simulations exhibited the greatest sensitivity to $N_{leaf}$ and $V_{cmax25}$, these parameters were also found to have the highest identifiability (Fig. 2a and Fig. 3). Given the high sensitivity of COS modeling to $V_{cmax25}$, $N_{leaf}$ and $b_{H_2O}$, it is unsurprising that these parameters can be effectively identified by the assimilation of COS. However, our findings indicate that the sensitivity of $V_{cmax25}$, $N_{leaf}$ is much greater than that of $b_{H_2O}$, yet the latter is much more identifiable. This outcome can be attributed to the highly significant correlation between $V_{cmax25}$ and $N_{leaf}$, as parameter interaction is a primary contributor to parameter unidentifiability (Gan et al., 2014).

In Sect. 3.1, it was demonstrated that the modeling of COS exhibits a low sensitivity to $f_{leaf}$, $m_{H_2O}$ and $VJ_{slope}$. Consequently, it is reasonable that the assimilation of COS did not effectively identify $f_{leaf}$, $m_{H_2O}$ and $VJ_{slope}$ (Fig. 3). However, due to their significant correlations with other plant growth-related parameters, effective identification is possible in specific cases. Notably, combinations such as $m_{H_2O}$ - $b_{H_2O}$ and $m_{H_2O}$ - $V_{cmax25}$ are very significantly correlated (Fig. 6b), and both $b_{H_2O}$ and $V_{cmax25}$ were identified at FI-Hyy. As a result, $m_{H_2O}$ also attain highly identifiability at this site.

It has been previously demonstrated that soil hydrology-related parameters exert a minimal impact on COS simulations (Fig. 2) and cannot be effectively constrained through COS assimilation in general (Fig. 3). Consequently, these parameters exhibit low identifiability, although significant combinations of correlations associated with soil hydrology-related parameters were observed at certain sites (e.g., DK-Sor).

## 4.4 Relationship between COS and GPP simulation performances

In this study, we identified the top 100 parameter sets, whose corresponding simulations displayed the smallest RMSE concerning COS observations, as the behavioral parameter sets. Subsequently, these behavioral parameter sets were employed to derive the posterior simulated COS and GPP, and to estimate prediction uncertainty. Therefore, it is necessary to investigate the distribution of RMSEs for COS simulations and GPP simulations, and to understand the relationship between the model performance of COS and that of GPP.

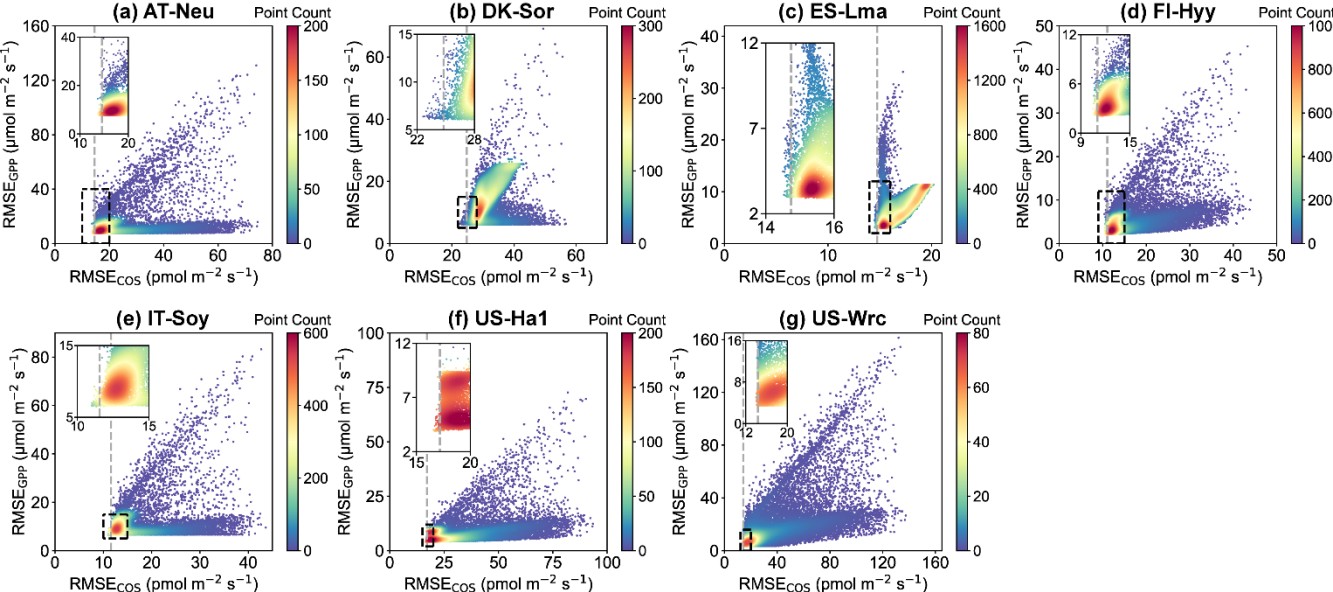

**Figure 7.** Comparison of RMSE for COS (RMSE_COS) and GPP (RMSE_GPP) in the Monte Carlo simulations. Each data point represents a parameter set, with color indicating data density. The gray dashed line represents the RMSE threshold for COS simulations, calculated as the mean of the 100th and 101st smallest values of the RMSE.

In Fig.7, scatter plots of RMSEs for COS and GPP are presented. It can be observed that at most sites, where the scatters are most densely distributed, there tend to be relatively small RMSEs for both COS and GPP. These results indicate that the current model is generally capable to simulate COS and GPP well at the same time. However, given the distinct mechanisms of COS and GPP as well as the uncertainties of model structure and driving data etc., there are also numerous parameter sets that perform well for either COS or GPP but exhibit significant discrepancies with the observations of the other. For example, the model runs with the 3 % highest RMSE for GPP instead exhibit good performance in terms of COS at ES-Lma, with their mean RMSE values (15.42 pmol m$^{-2}$ s$^{-1}$) less than that of the prior (17.10 pmol m$^{-2}$ s$^{-1}$). Overall, our results suggest that these behavioral parameter sets, which demonstrate good performance in COS simulation, also generally perform well in modeling GPP. However, the parameter sets with relatively good GPP simulation results exhibit significant variability in the performance of COS modeling.

## 4.5 Caveats and implication

Compared to the big leaf model, the two-leaf model has been demonstrated to better describe the canopy radiation distribution, GPP, and stomatal conductance (Luo et al., 2018). In this study, we take the advantage of two-leaf model to simulate COS fluxes from plant and soil based on the BEPS model within the two-leaf framework. Ecosystem COS flux data were used to calibrate the model parameters belong to BEPS and to optimize GPP simulations among diverse ecosystems within the Monte Carlo-based methodology. Our results demonstrate that COS not only improves the accuracy of GPP simulations but also reduces GPP simulation uncertainty. However, due to the lack of in-situ COS concentration and flux data, as well as BEPS model driving data (e.g., meteorological data, LAI data and clumping index data), we are still facing challenges in evaluating the performance of the two-leaf model compared to other models in COS simulation. issue. Therefore, there is an urgent need for more in situ meteorological data, vegetation canopy structural parameters, as well as COS observations.

The spatial and temporal variation in atmospheric COS concentrations has a considerable influence on the COS plant uptake (Ma et al., 2021; Kooijmans et al., 2021) due to the linear relationship between the two (Stimler et al., 2010). With the lack of continuous ground-based COS concentration observations, COS concentrations in the bulk air are assumed to be spatially invariant over the globe but to vary annually in this study, which may lead to significant biases in COS simulations. Currently, several recent studies have simulated COS vegetation fluxes based on atmospheric transport model-derived COS concentration data within the big-leaf framework (Kooijmans et al., 2021; Maignan et al., 2021; Abadie et al., 2023). These COS fluxes simulated based on big-leaf models were in turn used to drive atmospheric transport models (Remaud et al., 2023; Ma et al., 2023). Within an atmosphere inversion framework, recent studies indicate an underestimation of the biosphere COS sink in high-latitude regions of the Northern Hemisphere (NH) (Ma et al., 2021; Remaud et al., 2023). Larger underestimations of ecosystem COS exchange based on big-leaf model at high latitudes have also been confirmed at the site scale, and the underestimations of COS are consistent with biases in GPP for some sites (Kooijmans et al., 2021). Interestingly, Luo et al. (2018) demonstrated that the reason for the underestimation of GPP by the big-leaf model is that it fails to accurately describe the instantaneous radiation distribution in the canopy, and the underestimation increases with the increase of LAI. The NH

high-latitude regions have relatively high LAI (Fang et al., 2019), therefore the deficiency of the big leaf model in simulating radiation distribution may contribute to the existence of the missing COS sink in the NH high latitude in summer, and this deficiency is amplified by the larger LAI. In fact, the spatial distribution of LAI (i.e., GLOBMAP LAI) retrieved through remote sensing not only in NH high-latitude regions but also in central Africa aligns with the spatial distribution of the missing sink revealed by the "objective" inversion conducted by Ma et al. (2021) (as illustrated in Figure 7 in Ma et al. (2021)), which further validates the reasonableness of this hypothesis. Therefore, conducting COS simulations under the two-leaf framework at a global scale holds the promise of providing insights into the global COS vegetation sink and benefiting the simulation of the spatial and temporal distribution of COS concentrations. Thus, it is necessary to conduct regional and global COS simulations within the two-leaf model framework in the future.

Taking advantage of the Monte Carlo-based parameter optimization approach, we analyzed the global sensitivity, identifiability as well as interactions of COS-related parameters in this study. Furthermore, we quantified the uncertainty in simulated COS and GPP, thereby revealing the capacity of COS to constrain the uncertainty in GPP simulations. However, the Monte Carlo-based parameter optimization approach is subject to controversy (Sambridge and Mosegaard, 2002) due to the numerous subjective decisions involved in its implementation, such as the selection of parameter range, sample size and performance metric, etc. Further research is needed to investigate the impact of these factors on the parameter optimization results related to COS and the assessment of model prediction uncertainty.

## 5 Conclusions

In this study, carbonyl sulfide flux data were utilized to calibrate the ecosystem model parameters and to optimize GPP simulations among various ecosystems within a Monte Carlo-based approach using COS modeling within BEPS. A global parameter sensitivity analysis was conducted to identify the most sensitive ones among a set of 9 pre-selected parameters. The identifiability and interaction of model parameters were investigated by the behavioral parameter sets. The effectiveness of COS in improving the model performance of GPP was evaluated. The major findings are as follows:

(1) Similar to GPP, we found the modeling of COS is sensitive to parameters $V_{cmax25}$ and $N_{leaf}$, while insensitive to soil hydrology related parameters as well as the energy related parameter $f_{leaf}$. Unlike GPP, COS is sensitive to $b_{H_2O}$ while being insensitive to $m_{H_2O}$ and $VJ_{slope}$.

(2) The assimilation of COS within the Monte Carlo-based approach effectively improved model performance of GPP and significantly reduced the model predictive uncertainty, with a mean RMSE reduction of 40.56 % ± 13.77 % and a mean range width reduction as high as 65.81% ± 6.77 %.

(3) Complex and significant two-parameter or multi-parameter interactions exist between the model parameters. Particularly, $V_{cmax25}$ and $N_{leaf}$ show highly significant correlations (p < 0.01) at all sites.

(4) Generally, $b_{H_2O}$, $V_{cmax25}$ and $N_{leaf}$ can be well identified through the assimilation of COS, especially $b_{H_2O}$, whereas the soil hydrology related parameters and the light-reaction related parameters cannot be identified effectively.

## A1 BEPS photosynthesis and stomatal conductance modeling approach

In the BEPS model, the net photosynthesis rate (A) is calculated using the Farquhar model (Farquhar et al., 1980; Chen et al., 1999):

$$A = \min(A_i, A_j) - R_d \tag{A1}$$

$$A_c = V_{cmax} \frac{C_i - \Gamma_i^*}{C_i + K_c\left(1 + \frac{O_i}{K_o}\right)} \tag{A2}$$

$$A_j = J \frac{C_i - \Gamma_i^*}{4(C_i - 2\Gamma_i^*)} \tag{A3}$$

where $A_i$ and $A_j$ are Rubisco-limited and RuBP-limited gross photosynthetic rates ($\mu mol\,m^{-2}s^{-1}$), respectively. $R_d$ is leaf dark respiration ($\mu mol\,m^{-2}s^{-1}$). $V_{cmax}$ is the maximum carboxylation rate of Rubisco ($\mu mol\,m^{-2}s^{-1}$); J is the electron transport rate ($\mu mol\,m^{-2}s^{-1}$); Ci and Oi are the intercellular carbon dioxide ($CO_2$) and oxygen ($O_2$) concentrations ($mol\,mol^{-1}$), respectively; Kc and Ko are Michaelis–Menten constants for $CO_2$ and $O_2$ ($mol\,mol^{-1}$), respectively.

The electron transport rate, J, is dependent on incident photosynthetic photon flux density (PPFD, $\mu mol\,m^{-2}s^{-1}$) as:

$$J = \frac{J_{max}\,I}{I + 2.1 J_{max}} \tag{A4}$$

where $J_{max}$ is the maximum electron transport rate ($\mu mol\,m^{-2}s^{-1}$), $I$ is the incident PPFD calculated from the incident shortwave radiation $R_{SW}$ ($W\,m^{-2}$):

$$I = \beta\,R_{SW}\,f_{leaf} \tag{A5}$$

where $\beta = 4.55$ is the energy – quanta conversion factor ($\mu mol\,J^{-1}$), $f_{leaf}$ is the ratio of photosynthesis active radiation to the

605 shortwave radiation (unitless).

The maximum carboxylation rate of Rubisco $V_{cmax}$ was calculated according to the modified Arrhenius temperature function (Medlyn et al., 2002) and the maximum carboxylation rate of Rubisco at 25 °C ($V_{cmax25}$). $V_{cmax}$ is generally proportional to leaf nitrogen content. Considering both the fractions of sunlit and shaded leaf areas to the total leaf area and the leaf nitrogen content vary with the depth into the canopy, the $V_{cmax}$ values of sunlit ($V_{cmax,sunlit}$) and shaded ($V_{cmax,shaded}$)

leaves can be obtained through vertical integrations with respect to canopy depth (Chen et al., 2012; De Pury and Farquhar, 1997):

$$V_{cmax,sunlit} = V_{cmax}\chi_n N_{leaf} \frac{k[1 - e^{(k_n+k)L}]}{(k_n + k)(1 - e^{-kL})} \tag{A6}$$

$$V_{cmax,shaded} = V_{cmax}\chi_n N_{leaf} \frac{\frac{1}{k_n}[1 - e^{-k_n L}] - \frac{1}{k_n + k}[1 - e^{(k_n+k)L}]}{L - \frac{1}{k}(1 - e^{-kL})} \tag{A7}$$

where $\chi_n$ (m² g⁻¹) is the relative change of $V_{cmax}$ to leaf nitrogen content; $N_{leaf}$ (g m⁻²) is the leaf nitrogen content at the top of the canopy; $k_n$ is the leaf nitrogen content decay rate with increasing depth into the canopy, taken as 0.3; L is the canopy depth described in total LAI. $k$ is calculated as:

$$k = G(\theta)\Omega \cos(\theta) \tag{A8}$$

where G($\theta$) is the projection coefficient, taken as 0.5.

After $V_{cmax}$ values for the representative sunlit and shaded leaves are obtained, the maximum electronic transport rate for the sunlit and shaded leaves are obtained from Medlyn et al. (1999):

$$J_{max} = VJ_{slope} V_{cmax} - 14.2 \tag{A9}$$

where $VJ_{slope}$ (unitless) is the slope of the relationship of $V_{cmax}$ and $J_{max}$.

The leaf stomatal conductance to water vapor ($g_{sw}$ in mol m⁻²s⁻¹) is estimated using a modified version of Ball-Berry (BB) empirical model (Ball et al., 1987) following Woodward et al. (1995):

$$g_{sw} = b_{H_2O} + \frac{m_{H_2O} \ A \ R_h \ f_w}{C_a} \tag{A10}$$

where $b_{H_2O}$ is the intercept of the BB model, representing the minimum $g_{sw}$ (mol m⁻²s⁻¹), $m_{H_2O}$ is the empirical slope parameter in the BB model (unitless), $R_h$ is the relative humidity at the leaf surface (unitless), $f_w$ is a soil moisture stress factor describing the sensitivity of $g_{sw}$ to soil water availability (Ju et al., 2006), $C_a$ is the atmospheric CO₂ concentration (µmol mol⁻¹).

Soil water availability factor $f_{w,i}$ in each layer $i$ is calculated as:

$$f_{w,i} = \frac{1.0}{f_i(\psi_i)f_i(T_{s,i})} \tag{A11}$$

where $f_i(\psi_i)$ is a function of matrix suction $\psi_i$ (m) (Zierl, 2001), $f_i(T_{s,i})$ is a function describing the effect of soil temperature ($T_{s,i}$ in °C) on soil water uptake (Bonan, 1991).

To consider the variable soil water potential at different depths, the scheme of Ju et al. (2006) was employed to calculate the weight of each layer ($w_i$) to $f_w$:

$$w_i = \frac{R_i f_{w,i}}{\sum_{i=1}^{n} R_i f_{w,i}} \tag{A12}$$

where $n$ is the number of soil layers (five were used in this study) of the BEPS model, $R_i$ is the root fraction in layer $i$, calculated as:

$$R_i = \begin{cases} 1 - r_{decay}^{100cd_i} & i = 1 \\ r_{decay}^{100cd_{i-1}} - r_{decay}^{100cd_i} & 1 < i < n \\ r_{decay}^{100cd_{i-1}} & i = n \end{cases} \tag{A13}$$

where $cd_i$ is the cumulative depth (m) of layer $i$. In this study, each soil layer depth (from top to bottom) of the BEPS model is 0.05 m, 0.10 m, 0.20 m, 0.40 m and 1.25 m, respectively.

The overall soil water availability $f_w$ is then calculated as:

$$f_w = \sum_{i=1}^{n} f_{w,i} w_i \tag{A14}$$

The hydraulic conductivity of each soil layer $K_i$ (m s$^{-1}$) is expressed as:

$$K_i = Ksat_i \left(\frac{swc_i}{\theta_{s,i}}\right)^{2b_i+3} \tag{A15}$$

where $Ksat_i$ is the saturated hydrological conductivity of soil layer $i$ (m s$^{-1}$); $SWC_i$ is the volumetric liquid soil water content of soil layer $i$ (m s$^{-1}$); $\theta_{s,i}$ is the porosity of soil layer $i$ (unitless); $b_i$ is the Campbell parameter for soil layer $i$, determining the change rate of hydraulic conductivity with SWC (unitless). In this study, $Ksat_i$ and $b_i$ are expressed as:

$$Ksat_i = Ksat_{scalar} Ksat_{df,i} \tag{A16}$$

$$b_i = b_{scalar} b_{df,i} \tag{A17}$$

where $Ksat_{df,i}$ and $b_{df,i}$ are the default values of $Ksat_i$ and $b_i$ respectively.

**A2  BEPS soil COS modeling approach**

The total soil COS flux $F_{COS,soil}$ is the sum of abiotic COS flux $F_{COS,abiotic}$ and biotic COS flux $F_{COS,biotic}$.

$$F_{COS,soil} = F_{COS,abiotic} + F_{COS,biotic} \tag{A18}$$

Here, we take the approach developed in Whelan et al. (2016) for the modeling of $F_{COS,soil}$. In this approach, $F_{COS,abiotic}$ is described as an exponential function of the temperature of soil $T_{soil}$ (°C).

$$F_{COS,abiotic} = alpha\, e^{beta\, T_{soil}} \tag{A19}$$

where $alpha$ and $beta$ were parameters determined using the least-squares fitting approach. We assigned the values of $alpha$ and $beta$ to BEPS according to the parameterizations scheme of Whelan et al. (2016).

$F_{COS,biotic}$ is described as the product of a power function and an exponential function.

$$F_{COS,biotic} = F_{opt} \left(\frac{SWC}{SWC_{opt}}\right) e^{-a\left(\frac{SWC}{SWC_{opt}}-1\right)} \tag{A20}$$

$$a = ln\left(\frac{F_{opt}}{F_{SWC_g}}\right)\left(ln\left(\frac{SWC_{opt}}{SWC_g}\right) + \left(\frac{SWC_g}{SWC_{opt}} - 1\right)\right)^{-1} \tag{A21}$$

Here $a$ is the curve shape constant. The maximum biotic COS uptake $F_{opt}$ and the biotic COS uptake $F_{SWC_g}$ are the COS fluxes (pmol m$^{-2}$ s$^{-1}$) at optimum soil water content $SWC_{opt}$ and a secondary soil water content $SWC_g$, and $SWC_g > SWC_{opt}$. A more detailed description of the modeling of $F_{COS,biotic}$ and the parameterization scheme adopted in this study can be found in Whelan et al. (2022).

**Appendix B: Additional figure**

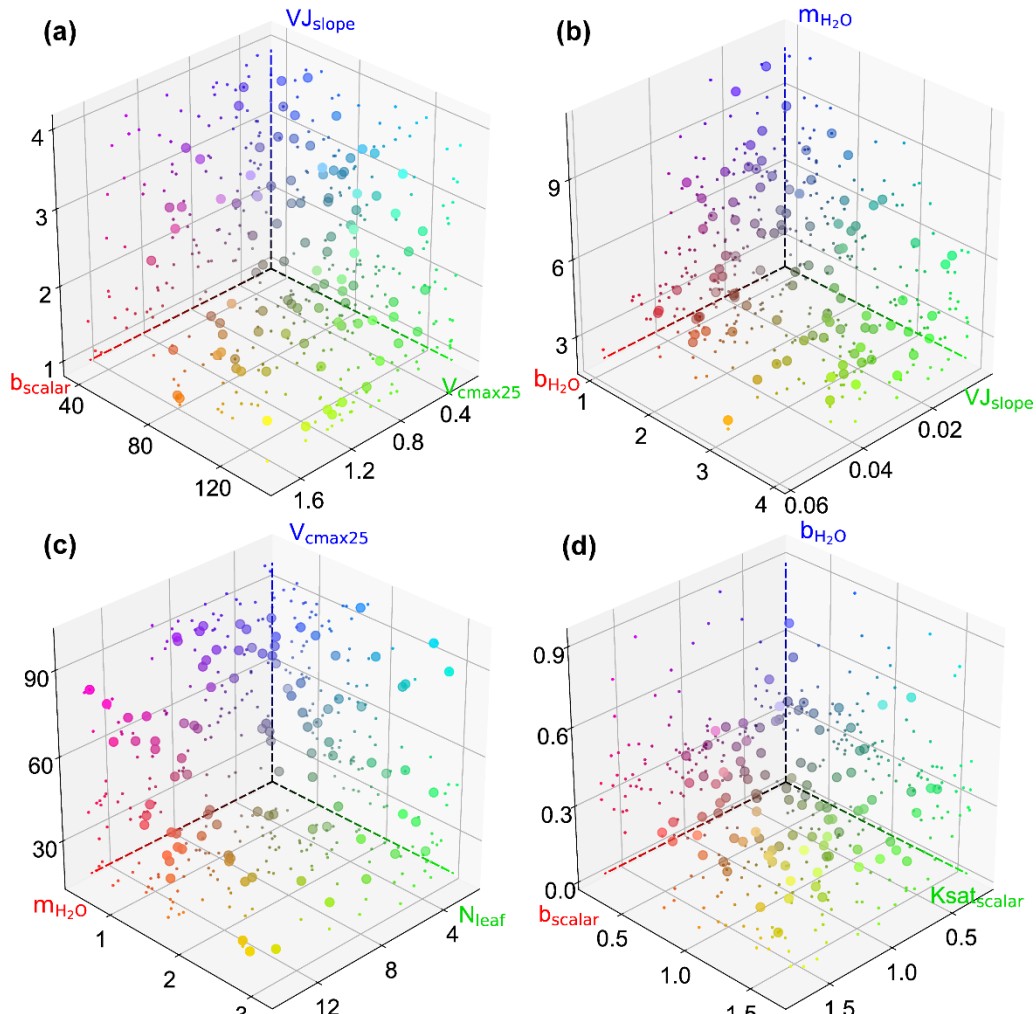

**Figure B1.** Scatter plots showing the behavioral parameter sets in 3D parameter space at AT-Neu (a), FI-Hyy (b) and US-Ha1 (c) and US-Wrc. The scatter colors represent the magnitude of the corresponding parameters using RGB values. The projection of the scatter is illustrated with smaller markers.

*Data availability.* Measured eddy covariance carbonyl sulfide fluxes data can be found at https://zenodo.org/record/3406990 for AT-Neu, DK-Sor, ES-Lma and IT-Soy, https://zenodo.org/record/6940750 for FI-Hyy, and from the Harvard Forest Data Archive under record HF214 (https://portal.edirepository.org/nis/mapbrowse?packageid=knb-lter-hfr.214.4 ) for US-Ha1.The raw COS concentration data of US-Wrc can be obtained at https://zenodo.org/record/1422820. The meteorological data can be obtained from the FLUXNET database (https://fluxnet.org/) for AT-Neu, DK-Sor, ES-LMa, FI-Hyy and US-Ha1; from the AmeriFlux database (https://ameriflux.lbl.gov/) for US-Ha1 (except shortwave radiation data) and US-Wrc; from the ERA5 dataset (https://cds.climate.copernicus.eu/cdsapp#!/dataset/reanalysis-era5-single-levels?tab=overview) for AT-Neu, IT-Soy

and US-Ha1. The GPP data can be obtained from the FLUXNET database for DK-Sor, ES-LMa, FI-Hyy and US-Ha1; from the AmeriFlux database for US-Ha1; from https://zenodo.org/record/6940750 for AT-Neu and IT-Soy; and from https://zenodo.org/record/1422820 for US-Wrc. The GLASS LAI is available at ftp://ftp.glcf.umd.edu/ and the GLOBMAP LAI is available at https://zenodo.org/record/4700264#.YzvSYnZBxD8%2F.

*Author contributions.*

MW designed the experiments and developed the model. XX improved the model and performed the Monte Carlo simulations. HZ made the analysis and wrote the original manuscript. All the authors contributed to the writing of the manuscript.

*Competing interests.* The authors declare that they have no conflict of interest.


*Acknowledgements.* This study was supported by the National Natural Science Foundation of China (42371486, 42111530184), the Research Funds for the Frontiers Science Center for Critical Earth Material Cycling, Nanjing University (Grant No: 090414380031, 020914380115). We acknowledge Prof. Tim Moore from McGill University to help us in improving the language.

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
