# Peer review of "Optimizing the terrestrial ecosystem gross primary productivity using carbonyl sulfide (COS) within a two-leaf modeling framework"

_EGUsphere, 2023_

## Referee Comment (RC1)

Review of egusphere-2023-3032 (referee comment)

**General comments**

Zhu et al. investigated the terrestrial GPP estimation using COS within a two-leaf modelling framework. COS flux data were used to calibrate the ecosystem model parameters and to optimize GPP simulations among different ecosystems within the Monte Carlo-based methodology base on the coupling of COS modeling and the BEPS model. The approach is with novelty, and brings new method and knowledge to the field of carbon cycle and also improves the estimation of GPP. In general, the work presented in the manuscript is interesting and worthy of publication. However, there are a few issues the authors should address before publication. The figures, tables and citations are not carefully maintained. The storyline is sometimes hard to follow.

**Specific comments**

COS fluxes measurements are used to assimilate and improve the BEPS model and GPP estimate. COS itself is also a trace gas in the atmosphere, and the authors are suggested to summarize the measurements of COS mole fractions and relevant modelling studies.

The manuscript is related to another manuscript, Zhu et al, (2023 under review). Authors are advised to cite the previous one and discuss relevance to the current manuscript, e.g. the model approach.

The discussion part is suggested to include a discussion of advantage and disadvantage of the model work.

**Technical corrections and Typing errors**
Line 1: The title "two-leaf" could be two-leaf without "".
Line 5: change to the affiliation only without currently at. If the co-author is currently only at this affiliation, please indicate with a superscript.
Line 18: 'two-leaf' to two-leaf, and elsewhere.
Line 19: "through the fusion of COS data" to "through the data assimilation of COS flux measurements".
Line 27: GPP should be one keyword. Model-data fusion is not accurate, use data assimilation.
Line 55: "not only the model variables like GPP are expected to be optimized" to "not only the model variables like GPP are expected to be improved".

Line 57: "through assimilating the COS data" to "through assimilating the COS flux measurements".
Line 58: here more related papers should be cited, e.g. Zhu et al., 2023.
Line 65: "LSM" to "a LSM".
Line 72-75: it is too vague to read. Please rewrite what you are going to do in details.
Line 90: "two-leaf" to two-leaf.
Line 95: the model description is not clear enough. Suggest move details to the main text from appendix A1.
Table 1: Is there missing data in a whole year? How do you deal with the missing data?
Section 2.3.1: how do you select the satellite LAI data to best match the field measurements?
Line 129: define ERA5.
Section 2.4: Is it the optimization approach? If so, please rename the section title to show the method explicitly.
Line 159: please refer to literatures.
Line 164: define "behavioral and non-behavioral simulations".
Line 207: "influence GPP modeling but have minimal impacts on COS modeling." To "influence GPP simulations but have minimal impacts on COS simulations."
Figure 1: the parameters need to be explained in the figure caption.
Line 220: Here the text refers to Fig. 2?
Figure 2: there are many subplots in the figure, maybe make it bigger.
Figure 2: explain the parameters and PI in the figure caption.
Line 281: here you refer to Fig. 3? Also Line 287-288.
Table 3: define reduction in percentage.
Figure 3: the order of numbering is something wrong. IT-Soy should be (d).
Figure 3: it is confusing that some panels have x-axis labeled as year, while others are labeled as Day of year. Please make it in consistency.
Line 334: refer to Fig. 4.
Line 341: refer to Fig. 4d.
Figure 4: IT-Soy should be (d).
Table 2 and Table 4: why is RMSE reduction of COS range width is much larger than that of GPP?
Figure 5 and Figure C1: move Figure C1 to main text. Or combine Figure 5 and Figure C1.
Line 390: "Knauer et al., 2020" is not in the Reference.
Line 394: "Ma et al., 2022" is not in the Reference. check reference if all of them are properly cited in the main text.
Line 420: remove "To provide deeper insights into these interactions and highlight significantly correlated parameter combinations, we generated

Fig. 6."

Line 421: "This figure …" To "Figure 6 …"

Figure 6: It is not easy to interpret the information from 3D view. Please try cross-section.

Line 436: define PI before using it.

Line 456: provide citation or the text you refer to.

Line 474: "show a significant range of variation", provide an estimate of the range.

Line 482: "COS modeling" to "COS simulation".

Reference:

Zhu, H., Wu, M., Jiang, F., Vossbeck, M., Kaminski, T., Xing, X., Wang, J., Ju, W., and Chen, J. M.: Assimilation of Carbonyl Sulfide (COS) fluxes within the adjoint-based data assimilation system–Nanjing University Carbon Assimilation System (NUCAS v1.0), EGUsphere [preprint], https://doi.org/10.5194/egusphere-2023-1955, 2023.

---

## Author Comment (AC1)

**Response to the comments of Reviewer #1**

We would like to thank the anonymous referee for his/her comprehensive review and valuable comments. These comments helped improve and clarify the submitted manuscript. In response, we have made changes according to the referee's suggestions. Below we reply to each comment point by point, showing the reviewers' comments in black and our responses in blue. Changes to the original manuscript are highlighted in **bold blue**. Note that the line numbers in the response are updated based on the revised manuscript, which we provide with our response.

**General comments**

Zhu et al. investigated the terrestrial GPP estimation using COS within a two-leaf modelling framework. COS flux data were used to calibrate the ecosystem model parameters and to optimize GPP simulations among different ecosystems within the Monte Carlo-based methodology base on the coupling of COS modeling and the BEPS model. The approach is with novelty, and brings new method and knowledge to the field of carbon cycle and also improves the estimation of GPP. In general, the work presented in the manuscript is interesting and worthy of publication. However, there are a few issues the authors should address before publication. The figures, tables and citations are not carefully maintained. The storyline is sometimes hard to follow.

Response: Thank you for your valuable feedback. We acknowledge the oversight in maintaining the figures, tables, and citations, as well as the shortcomings in ensuring the coherence of the storyline. In response, we have revised the inappropriate figures, tables, and citations in the original manuscript, and rewritted relevant sentences to enhance the coherence and readability of the article.

**Specific comments**

COS fluxes measurements are used to assimilate and improve the BEPS model and GPP estimate. COS itself is also a trace gas in the atmosphere, and the authors are suggested to summarize the measurements of COS mole fractions and relevant modelling studies. The manuscript is related to another manuscript, Zhu et al, (2023 under review). Authors are advised to cite the previous one and discuss relevance to the current manuscript, e.g. the model approach. The discussion part is suggested to include a discussion of advantage and disadvantage of the model work.

Response: Thanks for the valuable comments. As you mentioned, this manuscript is related to another manuscript, i.e., Zhu et al. (2023). Indeed, both of these works are implemented within the two-leaf model framework. The difference lies in the fact that in the other manuscript, our main objective is to introduce our newly developed adjoint-based assimilation system (Nanjing University Carbon Assimilation System, NUCAS v1.0), demonstrate the robustness of the assimilation system, and investigate the constraints of the tracer-gas, e.g. carbonyl sulfide (COS) on water, energy and carbon related parameters and processes. While COS assimilation has proven effective in constraining COS-related model parameters and optimizing GPP, there remains a gap of systematic understanding of the interaction, identifiability of the optimized model parameters from different processes as well as the ability of COS in reducing model

prediction uncertainty of GPP.

In this manuscript, we address the shortcomings of the adjoint-based sensitivity analysis, which is based on the Bayesian approach, by employing the Monte Carlo-based parameter calibration method, and conducted a global sensitivity analysis to provide general results over the entire parameter space. Notably, we also analyzed COS-related parameter interaction, identifiability, as well as the constraint ability of COS on GPP uncertainty.

As for the measurements and modelling studies of COS mole fractions, they are also integral parts of research in the COS field, closely related and crucial to this study. Consequently, we fully agree with your point that it is necessary to summarize and discuss the measurements and modelling studies of COS mole fractions.

In view of this, we have made modifications to three aspects of the manuscript: (1) Clarifying the advantages of the two-leaf model, and explaining why we did not conduct comparative analyses between the two-leaf model and other models (i.e., the big-leaf model). (2) Increasing the citation of another study; discussing the advantages and disadvantages of the Monte Carlo-based parameter optimization approach. (3) Adding a summary and discussion of the observed and simulated studies related to COS mole fraction. Details are as follows:

(1) In the introduction, we have added a paragraph to introduce the rationality of the two-leaf model and the necessity of applying two-leaf model in LSM, as shown below: "**Due to the dissimilar illumination conditions, there are the significant variability of leaf photosynthesis between sunlit and shaded leaves (Chen et al., 1999; Pignon et al., 2017; Wang et al., 2018; Bao et al., 2022). It is now clearly recognized that big-leaf models are conceptually flawed and practically inaccurate and sunlit-shaded leaf stratification is necessary to make accurate canopy-level photosynthesis estimation (Chen et al., 2012; Luo et al., 2018). Consequently, in the process-based LSM that simulates COS plant uptake and photosynthesis in a coupled manner (Ball et al., 1987; Berry et al., 2013), the application of the two-leaf model shows promise for providing precise simulation of plant COS uptake.**" (line 71-76)

In the new added section (**Section 4.5 Caveats and implication**), we have clarified the reason why we did not conduct comparative analyses between the two-leaf model and other models (i.e., the big-leaf model), as shown below: "**Compared to big leaf model, two-leaf model has been demonstrated to better describe the canopy radiation distribution, GPP, and stomatal conductance (Luo et al., 2018). In this study, we take the advantage of two-leaf model, to simulate COS fluxes from plant and soil based on the BEPS model within the two-leaf framework. Ecosystem-scale COS flux data were used to calibrate the model parameters belong to BEPS and to optimize GPP simulations among diverse ecosystems within the Monte Carlo-based methodology. Our results demonstrate that COS not only improves the accuracy of GPP simulations but also reduces GPP simulation uncertainty. However, due to the lack of in-situ COS concentration and flux data, as well as BEPS model driving data (e.g. meteorological data, LAI data and clumping index data), However, due to the lack of in-situ COS concentration and flux data, as well as BEPS model driving data (e.g., meteorological data, LAI data, and clumping index data), we are still facing challenges in evaluating the performance of the two-leaf model compared to**

**other models in COS simulation. The increasing availability of observational data holds promise for addressing this issue.**" (line 538-546)

(2) We have added an introduction to another manuscript in the introduction section: "Currently, several studies have been endeavored to refine the model parameters of LSMs through assimilating the COS data, and thereby optimized the modeling of water-carbon fluxes (Chen et al., 2023; Abadie et al., 2023; **Zhu et al., 2023**)" (line 63)

This manuscript shares similarities with another manuscript (Zhu et al., 2023) in that both involve optimizing GPP using COS. In the other manuscript, we constructed an adjoint-based Nanjing University Carbon Assimilation System (NUCAS) and thus assimilated COS within NUCAS. In contrast, we utilized a Monte Carlo-based method to assimilate COS in this manuscript. Here, we took the advantage the Monte Carlo-based parameter optimization method, analyzed the global sensitivity, identifiability as well as interactions of COS-related parameters. Furthermore, the model prediction uncertainty for COS and GPP were evaluated. Thus, we added a discussion regarding this in the new added Section (**Section 4.5 Caveats and implication**), as shown below: "**Taking advantage of the Monte Carlo-based parameter optimization approach, we analyzed the global sensitivity, identifiability as well as interactions of COS-related parameters in this study. Furthermore, we quantified the uncertainty in simulated COS and GPP, thereby revealing the capacity of COS to constrain the uncertainty in GPP simulations. However, the Monte Carlo-based parameter optimization approach subject to controversy (Sambridge and Mosegaard, 2002) due to the numerous subjective decisions involved in its implementation, such as the selection of parameter range, sample size and performance metric, etc. Further research is needed to investigate the impact of these factors on the parameter optimization results related to COS and the assessment of model prediction uncertainty.**" (line 569-575)

(3) With reference the observed and simulated studies related to COS mole fraction, a summary of the trace gas COS in the atmosphere is included in the introduction: "**Carbonyl sulfide (COS) is the most abundant sulfur-containing trace gas in the atmosphere with a lifetime of about 2 years (Montzka et al., 2007; Karu et al., 2023). The tropospheric atmospheric mole fraction of COS is approximately 500 parts per trillion (ppt), exhibiting a typical seasonal amplitude of ~ 100–200 ppt (Montzka et al., 2007; Ma et al., 2021; Hu et al., 2021; Remaud et al., 2022; Remaud et al., 2023; Ma et al., 2023).**" (line 37-40)

A discussion about the modeling of COS mole fraction as well as the COS vegetation sink were included in the new added Section (**Section 4.5 Caveats and implication**), as shown below: "**The spatial and temporal variation in atmospheric COS concentrations has a considerable influence on the COS plant uptake (Ma et al., 2021; Kooijmans et al., 2021) due to the linear relationship between the two (Stimler et al., 2010). With the lack of continuous ground-based COS concentration observations, COS concentrations in the bulk air are assumed to be spatially invariant over the globe but to vary annually in this study, which may lead to significant biases in COS simulations. Currently, several recent studies have simulated COS vegetation fluxes based on atmospheric transport model-derived COS concentration data within the big-leaf framework (Kooijmans et al., 2021; Maignan et al., 2021; Abadie et al., 2023). These COS fluxes simulated based on big-leaf**

models were in turn used to drive atmospheric transport models (Remaud et al., 2023; Ma et al., 2023). Within an atmosphere inversion framework, recent studies indicate an underestimation of the biosphere COS sink in high-latitude regions of the Northern Hemisphere (NH) (Ma et al., 2021; Remaud et al., 2023). Larger underestimations of ecosystem COS exchange based on big-leaf model at high latitudes have also been confirmed at the site scale, and the underestimations of COS are consistent with biases in GPP for some sites (Kooijmans et al., 2021). Interestingly, Luo et al. (2018) demonstrated that the reason for the underestimation of GPP by the big-leaf model is that it fails to accurately describe the instantaneous radiation distribution in the canopy, and the underestimation increases with the increase of LAI. As we all know, the NH high-latitude regions have relatively high LAI (Fang et al., 2019). Therefore, the deficiency of the big leaf model in simulating radiation distribution may contribute to the existence of the missing COS sink in the NH high latitude in summer, and this deficiency amplified by the larger LAI. In fact, the spatial distribution of LAI  (i.e., GLOBMAP LAI) retrieved through remote sensing not only in NH high-latitude regions but also in central Africa aligns with the spatial distribution of the missing sink revealed by the "objective" inversion conducted by Ma et al. (2021) (as illustrated in Figure 7 in Ma et al. (2021)), which further validate the reasonableness of this hypothesis. Therefore, conducting COS simulations under the two-leaf framework at a global scale holds the promise of providing insights into the global COS vegetation sink and benefiting the simulation of the spatial and temporal distribution of COS concentrations. Thus, it is necessary to conduct regional and global COS simulations within the two-leaf model framework in the future." (line 547-568)

**Technical corrections and Typing errors**

Response: We sincerely appreciate the careful review and detailed comment provided by the reviewer.

Line 1: The title "two-leaf" could be two-leaf without "".

Response: Corrected.

Line 5: change to the affiliation only without currently at. If the co-author is currently only at this affiliation, please indicate with a superscript.

Response: Thank you for your comment. We have modified the affiliation accordingly.

Line 18: 'two-leaf' to two-leaf, and elsewhere.

Response: Corrected.

Line 19: "through the fusion of COS data" to "through the data assimilation of COS flux measurements".

Response: Thanks for your comment. We have made modifications to the sentence accordingly.

Line 27: GPP should be one keyword. Model-data fusion is not accurate, use data assimilation.

Response: Thanks for your comment. We have replaced the key word "Model-data fusion" with

"data assimilation" and included "GPP" as a key word.

Line 55: "not only the model variables like GPP are expected to be optimized" to "not only the model variables like GPP are expected to be improved".

Response: Corrected.

Line 57: "through assimilating the COS data" to "through assimilating the COS flux measurements".

Response: Corrected.

Line 58: here more related papers should be cited, e.g. Zhu et al., 2023.

Response: Thank you for your comment. We have now added a citation to this manuscript.

Line 65: "LSM" to "a LSM".

Response: Corrected.

Line 72-75: it is too vague to read. Please rewrite what you are going to do in details.

Response: Thanks for your comment. We have revised the sentences as follows: "**To address these questions, we utilized ecosystem COS flux data to optimize GPP across various ecosystems based on the coupling of COS modeling with the two-leaf based Biosphere-atmosphere Exchange Process Simulator (BEPS). Through Monte Carlo simulations, we conducted a global parameter sensitivity analysis to explore the sensitivity of COS and GPP simulations to model parameters related not only to photosynthesis but also to water and energy. The interaction and identifiability of these parameters were quantified using Monte Carlo optimized parameter sets. Additionally, the effectiveness of COS in constraining model uncertainty in simulated COS and GPP are evaluated.**" (line 84-89)

Line 90: "two-leaf" to two-leaf.

Response: Corrected.

Line 95: the model description is not clear enough. Suggest move details to the main text from appendix A1.

Response: Thanks for your valuable suggestion. To facilitate readers' understanding of our two-leaf COS and GPP simulation framework, we have moved the relevant model descriptions from Appendix A1 and Appendix A2 to the main text.

Table 1: Is there missing data in a whole year? How do you deal with the missing data?

Response: Thanks for your comment. The hourly COS flux observational data from these sites exhibit varying degrees of gaps, as illustrated in Figure 4. In Figure 4, we use the size of scatter points to represent the number of COS observational data points, aiming to provide readers with a rough understanding of the time periods and extent of missingness in the COS data. Additionally, for the majority of sites (except FI-Hyy and US-Ha1), the COS observational data time series are less than one year, typically around one month. Therefore, we have assigned different x-axis labels for different sites in Figure 4.

For most sites (except FI-Hyy), only the measured COS flux data are available and we did not do anything to deal with the missing data. Following the recommendations regarding the standardized processing of eddy covariance flux measurements of COS (Kohonen et al., 2020), both measured and gap-filled COS flux of FI-Hyy are provided in Vesala et al. (2022), and the latter were utilized in this study. We have also clarified this in the revised manuscript: "**Specifically, following the recommendations regarding the standardized processing of eddy covariance flux measurements of COS by Kohonen et al. (2020), both the measured and gap-filled COS flux observations are provided in Vesala et al. (2022), and the latter were utilized in this study.**" (line 184-186)

Section 2.3.1: how do you select the satellite LAI data to best match the field measurements?

Response: Thanks for your comment, the publications listed in **Table 1** provided in-situ LAI information for these sites, for example, the mean LAI values ($m^2\ m^{-2}$) during the campaign of AT-Neu, DK-Sor, ES-Lma and IT-Soy are provided in Table S1 of the supplementary material in Spielmann et al. (2019). Such information provides us with a reference for selecting LAI products. Now, we have rewritten the sentence to provide specific explanations for the selection of LAI, as follows: "**With reference to the observed LAI at these sites (Wehr et al., 2017; Rastogi et al., 2018; Spielmann et al., 2019; Kohonen et al., 2022), we used GLOBMAP products to drive the BEPS model at most sites (5/7) due to its good agreement with the observed LAI. Specifically, as the GLOBMAP product had considerably underestimated LAI at DK-Sor and was not consistent with the vegetation phenology at ES-Lma during the measurement campaign, GLASS LAI was used at these two sites.** (line 163-167)

Line 129: define ERA5.

Response: Thanks for your comment. Now the definition of ERA5, i.e., **European Centre for Medium-Range Weather Forecasts (ECMWF) Reanalysis v5** is added. (line 174-175)

Section 2.4: Is it the optimization approach? If so, please rename the section title to show the method explicitly.

Response: Thanks for your valuable comment. We have renamed the section title as "**2.4 The Monte Carlo-based parameter optimization approach**". (line 193)

Line 159: please refer to literatures.

Response: Thanks for your comment. Now, we have supplemented the original sentence with additional references, and the revised sentence reads as: "The prior values and prior ranges for these parameters (**Table 2**) were chosen based on literature **(Jackson et al., 1996; Medlyn et al., 1999; Kattge et al., 2009; Miner et al., 2017; Ryu et al., 2018)** and default model settings." (line 219-220)

Line 164: define "behavioral and non-behavioral simulations".

Response: Thanks for your comment. We have rewritten the sentences to include a definition of behavioral and non-behavioral simulations. "**Subsequently, model realizations are grouped into behavioral and non-behavioral model runs and associated parameter sets based on the values of the single or multiple performance measures and the predefined**

**threshold value (Houska et al., 2014). The former describes acceptable model realizations conditioned on the available observational data (Blasone et al., 2008; Beven and Binley, 2014). The latter describes parameter sets that produce behavior inconsistent with observed behavior.**" (line 199-203)

Line 207: "influence GPP modeling but have minimal impacts on COS modeling." To "influence GPP simulations but have minimal impacts on COS simulations."

Response: Corrected.

Figure 1: the parameters need to be explained in the figure caption.

Response: Thank you for your comment. Now, we have provided detailed descriptions of the parameters before presenting the results, as shown in **Table 2**.

Line 220: Here the text refers to Fig. 2?

Response: Yeah, thanks for your comment. We have thoroughly reviewed the manuscript to ensure accurate referencing of figures and tables.

Figure 2: there are many subplots in the figure, maybe make it bigger.

Response: Thanks for your comment. We have made every effort to present the figure as clearly as possible.

Figure 2: explain the parameters and PI in the figure caption.

Response: Thank you for your comment. Now, we have provided detailed descriptions of the parameters before presenting the results, as shown in **Table 2**. Additionally, the definition of PI has been added in the figure caption.

Line 281: here you refer to Fig. 3? Also Line 287-288.

Response: Yeah, thanks for your comment. We have thoroughly reviewed the manuscript to ensure accurate referencing of figures and tables.

Table 3: define reduction in percentage.

Response: Thank you for your comment. The definition of reduction has been added in the legend, as follows: **The reduction (%) of $RMSE_{mean}$ and range width is calculated as (1-posterior / prior) * 100.** (line 381-382)

Figure 3: the order of numbering is something wrong. IT-Soy should be (d).

Response: Thanks for your comment. In **Table 1**, the site characteristics were listed in alphabetical order. We intend to present the results in the same order (alphabetical order) in other figures, including **Figure 3** and **Figure 4** of the original manuscript. Therefore, in **Figure 3** and **Figure 4**, the subplot corresponding to IT-Soy has been assigned the label (e) instead of (d) based on alphabetical order of the site names, even though it is placed above the subplot corresponding to FI-Hyy for compact arrangement of the subplots. Regarding this matter, we have clarified in the legends: **The subplot numbers are assigned based on the alphabetical order of the site names.** (line 388-389)

Figure 3: it is confusing that some panels have x-axis labeled as year, while others are labeled as Day of year. Please make it in consistency.

Response: Thanks for your comment. Here, x-axis labels are assigned to the subplots according to the duration of the corresponding COS observational data. For the majority of sites (except FI-Hyy and US-Ha1), the duration of COS observational data is only about one month. In contrast, multi-year COS observations are available at FI-Hyy and US-Ha1. We are keenly aware of the importance of maintaining consistency in the labels of each subplot. However, given the significant differences in the duration of COS observational data across these sites, different x-axis labels ("Year" and "Day of year") have been assigned to the subplots in Figure 3 and Figure 4 of the original manuscript.

Line 334: refer to Fig. 4.

Response: Corrected.

Line 341: refer to Fig. 4d.

Response: Corrected.

Figure 4: IT-Soy should be (d).

Response: Thanks for your comment. Regarding this matter, we have already provided detailed explanations earlier, please refer to the preceding sections. Also, we have clarified the assignment of the subplot numbers: "**The subplot numbers are assigned based on the alphabetical order of the site names.**" (line 420-421)

Table 2 and Table 4: why is RMSE reduction of COS range width is much larger than that of GPP?

Response: Thanks for your valuable comment. The reason for this phenomenon is the difference of the sensitivity of simulated COS and GPP to the model parameters. Specifically, the parameters $m_{H_2O}$ and $VJ_{slope}$ strongly influence GPP modeling but have minimal impacts on COS modeling. Therefore, even after COS assimilation, these two parameters still have a wide posterior range, thus resulting in a large posterior range for GPP simulation. On the contrary, parameters that are sensitive to COS (i.e., those that have a significant impact on the posterior range of COS simulation) are well-constrained. As a result, there is a considerable reduction in the range of COS simulation.

Figure 5 and Figure C1: move Figure C1 to main text. Or combine Figure 5 and Figure C1.

Response: Thanks for your comment. According to your suggestion, we have combined these two figures.

Line 390: "Knauer et al., 2020" is not in the Reference.

Response: Corrected.

Line 394: "Ma et al., 2022" is not in the Reference. check reference if all of them are properly cited in the main text.

Response: Thanks for your comment. We have thoroughly reviewed the manuscript to ensure

precise citation.

Line 420: remove "To provide deeper insights into these interactions and highlight significantly correlated parameter combinations, we generated Fig. 6."

Response: Thank you for your comment. We have modified the sentence accordingly.

Line 421: "This figure …" To "Figure 6 …"

Response: Corrected.

Figure 6: It is not easy to interpret the information from 3D view. Please try cross-section.

Response: Thanks for your comment. The design of this figure is inspired by Figure 4 from Beven and Binley (2014). Similar to Beven and Binley (2014), we employ 3D plots to further explore, visually, the parameter space. The difference lies in the fact that Beven and Binley (2014) uses parameter likelihood thresholds to identify behavioral parameter sets and plots likelihood threshold surfaces in parameter space. In this study, we use an acceptable sampling rate to identify behavioral parameter sets. Thus, what is depicted here are collections of behavioral parameter sets in parameter space. However, fundamentally, our goal is to explore, visually, the parameter space, akin to Beven and Binley (2014). Therefore, following the suggestion of the reviewer #2, we remained this figure but relocated it to the appendix.

Line 436: define PI before using it.

Response: Thanks for your comment. We have already defined PI in Section 2.7. (line 253)

Line 456: provide citation or the text you refer to.

Response: Done.

Line 474: "show a significant range of variation", provide an estimate of the range.

Response: Thanks for your comment. In order to quantitatively describe the results, we revised the sentence as follows: **However, given the uncertainties of model parameters, structure and driving data etc., instances like at ES-Lma arise where the model runs with the 3 % highest RMSE for GPP instead exhibit good performance in terms of COS, with their RMSE values for COS all falling within the top 55 %. (line 526-528)**

Line 482: "COS modeling" to "COS simulation"

Response: Corrected.

***Reference:***

Zhu, H., Wu, M., Jiang, F., Vossbeck, M., Kaminski, T., Xing, X., Wang, J., Ju, W., and Chen, J. M.: Assimilation of Carbonyl Sulfide (COS) fluxes within the adjoint-based data assimilation system–Nanjing University Carbon Assimilation System (NUCAS v1.0), EGUsphere [preprint], https://doi.org/10.5194/egusphere-2023-1955, 2023

Response: Thank you for your suggestion regarding citing the manuscript. Now, we have already included a citation to this manuscript.

[revised manuscript text omitted]

---

## Author Comment (AC2)

**Response to the comments of Reviewer #2**

The authors have modelled vegetation and soil COS fluxes within the "two-leaf" version of the BEPS (Boreal Ecosystem Productivity Simulator) model. Then, they used observations of COS fluxes at seven sites and a Monte-Carlo approach to reduce parameter uncertainty in BEPS. They further evaluate the impact on GPP, and discuss parameter identifiability.

The paper is well built and very neat, the results are clearly presented. Some further explanations are however needed, and a few outlooks would be welcome.

We are truly grateful to the positive comments and thoughtful suggestions. These comments are all valuable and very helpful for revising and improving our manuscript. In response, we have made changes according to the referee's suggestions. Below we reply to each comment point by point, showing the reviewers' comments in black and our responses in blue. Changes to the original manuscript are highlighted in **bold blue**. Note that the line numbers in the response are updated based on the revised manuscript, which we provide with our response.

**Main comments**

*Abstract*

L14-15: "However, most of the current modeling approaches for COS and $CO_2$ did not explicitly consider the vegetation structural impacts, i.e. the differences between the sun-shade and sunlit leaves in COS uptake" -> It is a bit misleading that the authors bring forward such an argument, because they did not demonstrate in this paper the advantage of distinguishing between sunlit and shaded leaves. Why did not they show the impact of having a two-leaf model compared to a one flux model, as they were the first ones (to my knowledge) to use such a model? This would indeed have been a great achievement.

Response: Thanks for the valuable comments. In order to quantify the effect of changes in the quality of incoming radiation on photosynthesis, land surface models (LSMs) need to stratify the canopy into sunlit and shaded leaves and consider the differences in the transfer of direct and diffuse beams within the canopy (Mercado et al., 2009; He et al., 2013). The advantage of distinguishing between sunlit and shaded leaves in LSMs have been demonstrated in a number of studies (Wang and Leuning, 1998; Luo et al., 2018; Guan et al., 2022; Bao et al., 2022). Specifically, the performance of BL (big leaf), TBL (two big leaf), and TL (two leaf) upscaling scheme in estimating Evapotranspiration (ET) and gross primary productivity (GPP) using the Biosphere-atmosphere Exchange Process Simulator (BEPS) are evaluated with flux measurements from nine eddy covariance towers in Luo et al. (2018). They demonstrated that BL underestimates ET and GPP across all sites because the radiation gradient calculated based on Beer's law fails to describe the instantaneous radiation distribution in the canopy. As most current process-based plant COS uptake simulations are predominantly based on the Berry's stomatal conductance model of COS and the Ball-Barry model, the underestimation of GPP by the BL model ultimately impacts plant COS flux simulations.

As the advantages of the two-leaf model over both the big-leaf model and the two big-leaf model in terms of canopy radiation distribution, GPP, and stomatal conductance have been extensively discussed, we adopted the two-leaf model to simulate COS in this study. The reason

for not further comparing the results of the two-leaf COS model with those of other models based on COS observations is primarily twofold: the lack of accurate BEPS model driving data, and the absence of in-situ COS concentration and flux observation data.

(1) The lack of accurate BEPS model driving data. For sunlit and shaded leaf stratification, we need accurate description of the canopy structure with at least two structural parameters (Chen et al., 2012). One is the leaf area index (LAI), defined as one half the total (all sided) leaf area per unit ground surface area (Chen and Black, 1992). The other is the foliage clumping index characterizing the way that leaves in a canopy are spatially organized. Thus, In Luo et al. (2018), nine sites in Canada are selected mainly because they have some measured LAI, clumping index, and soil moisture data. The measured soil moisture data were utilized in Luo et al. (2018) to minimize the possible deviations in stomatal conductance modeling caused by the soil moisture simulation. Unfortunately, among the seven sites in this study, no continuous in-situ LAI or clumping index data were provided along with the COS data (Wehr et al., 2017; Rastogi et al., 2018; Spielmann et al., 2019; Vesala et al., 2022). The measured soil moisture data were also not available at US-Ha1. Furthermore, as mentioned in the manuscript, even continuous in-situ meteorological data were lacking at the IT-Soy site.

(2) The lack of COS concentration and flux observation data. Unlike $CO_2$, the concentration of COS exhibits strong seasonal variations, with seasonal amplitudes reaching up to 100-200 parts per trillion (Montzka et al., 2007; Kooijmans et al., 2021; Hu et al., 2021; Ma et al., 2021). Given the linear relationship between plant COS uptake and COS concentration (Stimler et al., 2010), these variations can significantly impact the simulation of COS plant fluxes. Unfortunately, continuous in-site COS concentration data are lacking at the sites. Moreover, for the majority (5/7) of sites in this study, the COS observation sequences are very short, lasting only about one month. As a contrast, observed sequences of GPP and ET at measurement sites all span over five years in Luo et al. (2018).

Overall, due to the lack of accurate BEPS model driving data and continuous in-site COS concentration data, the simulated COS flux subject to great uncertainty, whether it is based on the two-leaf model or other models. Furthermore, the majority of sites used in this study lack long time series of COS observations, and COS flux observations also exhibit considerable uncertainty (Kohonen et al., 2020). Therefore, we have refrained from comparing the COS simulation performance of the two-leaf model and other models. However, we do agree with your opinion and we also believe that comparing the COS simulation performance of the two-leaf model with other models (i.e., BL and TBL model) is an objective we should pursue, given the conceptual scientificity and practical robustness of the two-leaf model (Chen et al., 2012). We anticipate that the simulation of plant COS uptake based on the two-leaf model will outperform other models (i.e., BL and TBL model), and the global vegetation COS flux estimated based on the two-leaf model will exceed that estimated by other models. This will provide insights into both the accurate simulation of plant COS uptake and the magnitude and distribution of global COS vegetation sink.

**2.4.1 Parameter selection and sampling strategy**

L154-155: "9 parameters were selected to be calibrated in this study" -> Why didn't the authors perform a sensitivity analysis to select the most important parameters for COS and GPP? We are left with the impression that the selection was arbitrary, and we may fear that they have missed some important parameter.

Response: Thanks for the valuable comments. As mentioned in the manuscript, currently, numerous studies on parameter sensitivity in COS and GPP simulations have been conducted, laying the foundation for the parameter selection in this study. Specifically, the Morris method and RS-HDMR method were employed to identified that the sensitive parameters in simulating GPP by BEPS for 10 sites covering 7 plant functional types (PFT) over China in Xing et al. (2023). In this study, 21 model parameters were screened, encompassing not only photosynthesis-related parameters but also those associated with energy and water balance, heterotrophic respiration, and autotrophic respiration. The results highlighted that $V_{cmax25}$, , $N_{leaf}$, $r_{decay}$, $b_{H_2O}$, $m_{H_2O}$ and $f_{leaf}$ as the most crucial parameters for GPP simulation by BEPS. In another related manuscript (Zhu et al., 2023), we identified the model parameters sensitive to COS for BEPS. Therefore, the 9 parameters were selected to be calibrated in this study. Certainly, other literature listed in **Section 2.4.1** also provided references for our parameter selection.

We would like to highlight that the references listed in Section 2.4.1 have been updated. Specifically, the recently manuscript by **Abadie et al. (2023)** and **Zhu et al. (2023)** have been incorporated into the section. (line 216-217)

L155: Table B1 should be placed in the main manuscript, it's important to see here the detailed description of the parameters.

Response: Thanks for the comments. We have moved the Table into the main manuscript (renamed as **Table 2**).

**2.4.2 Selection of behavioral simulations**

"Behavioral simulation" is not an expression I've seen before. Could the authors use simpler terms like "selected" and "rejected" (for "non-behavioral")?

Thanks for your comment. The terms "behavioral" and "non-behavioral" have been extensively employed in the domain of Monte Carlo-based calibration, as evidenced by Beven and Binley (1992), Beven and Freer (2001) and Houska et al. (2014). Hence, we have maintained the usage of "behavioral" and "non-behavioral" in this context. In response, we have added introductions for "behavioral parameter sets" and "non-behavioral parameter sets". "**Subsequently, model realizations are grouped into behavioral and non-behavioral model runs and associated parameter sets based on the values of the single or multiple performance measures and the predefined threshold value (Houska et al., 2014). The former describes acceptable model realizations conditioned on the available observational data (Blasone et al., 2008; Beven and Binley, 2014). The latter describes parameter sets that produce behavior inconsistent with observed behavior.**"(line 199-203).

L168-169: "Thus, the deterministic model prediction is given by the ensemble mean of the 100 behavioral simulations." -> The authors could explain that the "100" comes from 0.5%*20,000.

Response: Thank you for your comments. Now we rewrite this sentence: "**Specifically, here we chose an ASR of 0.5%, i.e., the top 100 model runs with the lowest RMSE values for COS as behavioral simulations**." (line 234-235)

**2.6 Parameter uncertainty**

L183: "Due to the complexity of ecosystem" -> Could the authors be more specific: "Due to the functional and structural complexity of ecosystems"?

Response: Thanks for your comment. We have made the modification to the sentence accordingly. (line 247)

L193-194: "Taking into account the influence of the prior distribution to the behavioral parameter sets, the PI is defined as the reduction of the parameter range width. -> This means that if the initial range is overestimated, the PI may be artificially high. This could be the case for the $b_{H2O}$ parameter, where the max value (1) is 57 times larger than the initial value. Plus, the authors later write, citing Miner et al. (2017), that "83 % of the $b_{H2O}$ values are located between 0 and 0.15 mol m$^{-2}$ s$^{-1}$, and about half are located between 0 and 0.04 mol m$^{-2}$ s$^{-1}$" (L236-237).

Response: Thank you for your comments. As we mentioned in the manuscript, the default values and prior ranges for these selected parameters were chosen based on literature and default model settings. For $b_{H_2O}$, the default value of it in BEPS is 0.0175 mol m$^{-2}$ s$^{-1}$, and we assigned the prior range of it according to Miner et al. (2017). We also highlighted that "literature-documented values of $b_{H_2O}$ are highly variable". Actually, in the compilation provided by Miner et al. (2017), a number of documented values of $b_{H_2O}$ are already several tens of times greater than the prior value, for example, reaching as high as 0.57 mol m$^{-2}$ s$^{-1}$ in Bunce (2004) and 0.69 mol m$^{-2}$ s$^{-1}$ in Leuning (1995). Specifically, the value of 0.69 mol m$^{-2}$ s$^{-1}$ was provided alongside a corresponding standard deviation of 0.10 mol m$^{-2}$ s$^{-1}$. Considering the wide range of literature values of $b_{H_2O}$, we thus opted for a broad prior range (0-1 mol m$^{-2}$ s$^{-1}$) and performed the Monte Carlo simulations. Certainly, we acknowledge that the setting of prior ranges for parameters involves subjective decisions, and the prior range of $b_{H_2O}$ may be overestimated. Indeed, the involvement of subjective decisions is the primary reason for the controversy surrounding GLUE (Beven and Binley, 2014). In response, we have provided clarification regarding the subjectivity controversy surrounding Monte Carlo-based model calibration method. "**However, the Monte Carlo-based parameter optimization approach subject to controversy (Sambridge and Mosegaard, 2002) due to the numerous subjective decisions involved in its implementation, such as the selection of parameter range, sample size and performance metric, etc. Further research is needed to investigate the impact of these factors on the parameter optimization results related to COS and the assessment of model prediction uncertainty.**" (line 571-575)

**3.2 Posterior parameter distributions**

L252: The authors should explain what they call "the grouping value".

Response: Thank you for your comments. In Miner et al. (2017), the literature-documented values of $m_{H_2O}$ were grouped by plant function type (PFT). Thus, we reorganized the sentences as: "**Nevertheless, the optimization of $m_{H_2O}$ is generally achievable through COS assimilation, as supported by our results in good agreement with the compilation of Miner et al. (2017), in which the average historical values of $m_{H_2O}$ grouped by PFT (referred to as the PFT-grouping values below) are provided.**" (line 314-316).

Figure2. The authors should add 'COS' somewhere in the legend, document the boxplot (say it describes the posterior distribution), and explain axes, colours, title (PI).

Response: Thanks for your comment. We have revised the legend, as follows: "**Figure 3. Cumulative frequency distributions and boxplots for the posterior model parameters obtained by COS assimilation. The grey area represents uniform parameter distributions, while the colored areas denote posterior CDF distributions, with parameters for different sites represented using different colors. The box extends from the first quartile to the third quartile of the parameter values, with a line at the median. "×" markers denote outliers, and the whiskers represent the lowest or highest parameter values excluding any outliers. The black square represents the prior parameter value, and the axis ranges denote the prior ranges of the parameters. PI denote parameter identifiability, defined as the reduction of the parameter range width.**" (line 333-348)

**3.3 The optimization performance in COS fluxes**

L300-301: "despite remarkable improvement is attached by the posterior simulations" -> This is a weird formulation, to be rephrased.

Response: Thanks for your comment. The revised sentence reads as: "**Particularly, significant underestimation is found in the posterior simulations in 2017 for FI-Hyy, despite the posterior simulations shows a remarkable improvement in reproducing COS fluxes over the entire period (2013-2017).**" (line 364-376)

Figure 3/Figure 4: "The means and uncertainties of these observations and simulations are calculated and plotted on a daily or monthly scale" -> Do the authors compute the standard deviation of hourly values for daily means and over daily means for monthly means? Do they compute the standard error of the mean (SEM), defined as the standard deviation (SD) divided by the square root of the number of observations, and which would be more appropriate than SD to estimate the uncertainty of the mean?

Response: Thanks for your comments. Here the standard deviation of hourly values for daily means or monthly means were calculated.

The standard error of the mean (SEM) quantifies uncertainty in the estimate of the mean (Barde and Barde, 2012). However, our intention here is to quantify the uncertainties of the hourly observations on a daily or monthly scale, which does not align with the definition of SEM. Certainly, standard deviation (SD) quantifies the variability, which is also distinct with

uncertainty. Therefore, we have modified the corresponding sentence to clarify this. "The mean observed COS and its uncertainty **(estimated by the standard deviation)** are represented by black dots with error bars. The means and uncertainties of these **hourly** observations and simulations are calculated and plotted on a daily or monthly scale." (line 386-388 and line 418-420)

*4.2 Parameter interactions*

L407: "their weak equivalence" -> What do the authors mean? Equivalence to what?

Response: Thanks for your comment. The sentence is unnecessary, and we have deleted it.

Figure 6 is a bit difficult to interpret, I'm not sure it brings something, could it be moved to the Supplementary part?

Response: Thanks for your comments. The design of this figure is inspired by Figure 4 from Beven and Binley (2014). Similar to Beven and Binley (2014), we employ 3D plots to further explore, visually, the parameter space. Following your suggestion, we have relocated this figure to the appendix.

*4.3 Parameter identifiability*

L442: "the sensitivity of the input data to the parameter" -> This should rather be "the sensitivity of the modeled output to the parameter".

Response: Thanks for your comments. As you mentioned, it should rather be "the sensitivity of the modeled output to the parameter". More specifically, "modeled output" here refers to COS simulation. Therefore, we have revised the original sentence accordingly, and the modified sentence is as follows: "**In this study, the identifiability of a parameter closely related to the sensitivity of COS simulations to the parameter, although it is known to be influenced by model over-parameterization and parameter interactions (Gan et al., 2014).**" (line 497-498)

L446-447: "However, our findings indicate that the sensitivity of $Vc$max25, $Nl$eaf is much greater than that of $bH2O$, yet the latter is much more identifiable" -> An alternative explanation is once again the overestimated prior range of $bH2O$.

Response: Thanks for your comment. We acknowledge that the prior range of $b_{H_2O}$ may be overestimated and the overestimation of the prior range of $b_{H_2O}$ can be an alternative explanation of $b_{H_2O}$ being more identifiable (having larger PIs) as PI is defined as the reduction of the parameter range width. A detailed explanation of why we chose such a broad prior range for $b_{H_2O}$ has been provided previously, along with clarification of the drawback (i.e., involving subjective decisions) of the Monte Carlo-based parameter optimization approach.

L448: "as parameter interaction is a primary contributor to parameter unidentifiability" -> But then, this should also apply to $bH2O$, as it is highly correlated to fleaf and mH2O, as shown in Figure 5.

Response: Thanks for your comment. As shown in **Figure 5** and **Figure C1** of the original manuscript, there are complex correlations between the 9 pre-selected parameters, and $b_{H_2O}$ is indeed highly correlated to $f_{leaf}$ and $m_{H_2O}$ at AT-Neu. But as mentioned in the original manuscript, $N_{leaf}$ and $VJ_{slope}$ is the only parameter combination that is significantly correlated at all sites.

L456-457: "It has been previously demonstrated that soil hydrology-related parameters exert a minimal impact on COS simulations and cannot be effectively constrained through COS assimilation" -> That would depend on whether soil water stress conditions are present or not.

Response: Thanks for your comment. We agree with your point that the impact of soil hydrology-related parameters on COS simulations may vary depending on the presence of soil water stress conditions. We have revised the sentence as follows: "**It has been previously demonstrated that soil hydrology-related parameters exert a minimal impact on COS simulations (Figure 2) and cannot be effectively constrained through COS assimilation in general (Figure 3)**" (line 514-515)

*4.4 Relationship between COS and GPP simulation performance -> performances*

Response: Corrected

L464: "respond to RMSE" -> This seems awkward, to be rephrased.

Response: Thank you for your comment. We have revised the sentence as follows: "**Therefore, it is necessary to investigate the distribution of RMSEs for COS simulations and GPP simulations, and to understand the relationship between the model performance of COS and that of GPP.**" (line 517-519)

Figure 7: "Each data point represents a parameter set, with color indicating data density" -> That does not seem possible, some binning has to be made to get a density.

Response: Thank you for your comment. At each site, we actually obtained 20,000 discrete points distributed in a two-dimensional space of RMSE for COS ($RMSE_{COS}$) and GPP ($RMSE_{GPP}$), and some binning has to be made to get a density. In this study, we utilized kernel density estimation (https://docs.scipy.org/doc/scipy/reference/generated/scipy.stats.gaussian_kde.html) to estimate the probability density of each scatter point. Subsequently, we assigned colors to each scatter based on the estimated density, and plotted the scatter plots.

*5 Conclusions*

L485-486: "within the Monte Carlo-based methodology base on the coupling of COS modeling and the BEPS model" -> "with a Monte Carlo approach using COS modeling within BEPS"

Response: Thanks for your comment. We have revised the sentence accordingly. (line 577-578)

L486-487: "Global parameter sensitivity analysis was conducted to identify the sensitive parameters" -> "A global parameter sensitivity analysis was conducted to identify the most sensitive ones among a set of 9 pre-selected parameters."

Response: Thanks for your comment. We have revised the sentence accordingly. (line 578-579)

The conclusion is a bit abrupt. The authors should develop some outlooks. What are the consequences of this study? Is there a need to acquire more COS fluxes observations, or a need to improve the COS vegetation model? What will be the next steps with BEPS?

Response: Thank you for your valuable comment. As you mentioned, conducting more observations, developing advanced COS models (as done by Cho et al. (2023)), and utilizing varying COS concentrations for COS simulations are indeed our goals. Inevitably, this study was constrained by these factors. Regarding this, we have added a new section (**Section 4.5 Caveats and implication**) to discuss these issues and provide an outlook for our future work. Specifically, we have conducted COS simulations based on the two-leaf model at the site scale and utilized COS to optimized GPP. However, at the global scale, the scientific community is grappling with the COS missing sink issue, and the two-leaf model holds promise for addressing this problem. Thus, global COS simulations within two-leaf model are the next step awaiting our investigation. For a more detailed discussion on this aspect, please refer to **Section 4.5** in the revised manuscript.

*A2 BEPS leaf COS modeling approach*

L568: "where COS$a$ represents the COS mole fraction in the bulk air" -> Did the authors use a variable atmospheric COS mole fraction as it has been shown important (Kooijmans et al., 2021; Abadie et al., 2022)?

Response: Thank you for your valuable comment. We have revised the manuscript to include a more detailed description of the data used in this study. The revised sentence reads as follows: "**Data used in this study include LAI, land cover type, meteorological and soil data, as well as CO$_2$ and COS mole fraction data. The CO$_2$ and COS mole fractions in the bulk air were assumed to be spatially invariant over the globe but to vary annually. The CO$_2$ mole fraction data in this study are taken from the Global Monitoring Laboratory (https://gml.noaa.gov/ccgg/trends/global.html). For the COS mole fraction, we utilized the average of observations from sites SPO (South Pole) and MLO (Mauna Loa, United States) to drive the model. These data are publicly available online at: https://gml.noaa.gov/hats/gases/OCS.html.**" (line 153-158)

L572: How did the authors derive the empirical relationship expressed in equation (A19)?

Response: Thanks for your comment. Here we adapted the COS leaf uptake modeling approach from SiB4 (Equation 175 in Haynes et al. (2020)). Now, a more detailed description of the modeling approach is provided in the main manuscript (line 122-139):

**The leaf-level COS uptake rate $F_{cos,leaf}$ is determined by the formula (Berry et al., 2013):**

$$F_{cos,leaf} = COS_a \left( \frac{1.94}{g_{sw}} + \frac{1.56}{g_{bw}} + \frac{1}{g_{cos}} \right)^{-1} \tag{5}$$

**where $COS_a$ represents the COS mole fraction in the bulk air. $g_{sw}$ and $g_{bw}$ are the stomatal conductance and leaf laminar boundary layer conductance to water vapor (H$_2$O). The factors 1.94 and 1.56 account for the smaller diffusivity of COS with respect to H$_2$O. $g_{cos}$ indicates the apparent conductance for COS uptake from the intercellular airspaces, which combined the mesophyll conductance (Evans et al., 1994) and the biochemical**

reaction rate of COS and carbonic anhydrase (Badger and Price, 1994). It can be calculated as :

$$g_{COS} = \alpha V_{cmax} \tag{6}$$

Where $\alpha$ is a parameter that is calibrated to observations of simultaneous measurements of COS and $CO_2$ uptake (Stimler et al., 2012). $V_{cmax}$ is the maximum carboxylation rate of Rubisco. Analysis of these measurements yield estimates of α of ~1400 for C3 and ~7500 for C4 species. With reference the COS modelling scheme of the Simple biosphere model (version 4.2) (Haynes et al., 2020), $g_{cos}$ can be calculated as

$$g_{COS} = 1.4 * 10^3 * (1.0 + 5.33 * F_{C4}) * 10^{-6} F_{APAR} f_w V_{cmax} \tag{7}$$

where $F_{C4}$ denotes the C4 plant flag, taking the value of 1 for C4 plants and 0 otherwise. $f_w$ is a soil moisture stress factor describing the sensitivity of $g_{sw}$ to soil water availability (Ju et al., 2006). $F_{APAR}$ is the scaling factor for leaf radiation (Smith et al., 2008), calculated as:

$$F_{APAR} = 1 - e^{(-0.45\ LAI)} \tag{8}$$

**Minor comments**

L15: "i.e." -> "i.e.,"

Response: Corrected.

L15: I could not find information on the "sun-shade" expression, would not the simpler "shaded" be more appropriate?

Response: Thanks for your comment. Here "shaded" indeed is more appropriate and we have revised the sentence.

L31: "(GPP), is" -> "(GPP) is"

Response: Corrected.

L33: "the modeling of GPP are affected" -> "the modeling of GPP is affected"

Response: Corrected.

L60: "Ecosystem carbon, water and energy processes are interacted" -> "Ecosystem carbon, water and energy processes are interacting"

Response: Corrected.

L64: "e.g." -> "e.g.,"

Response: Corrected.

L67: "Which parameters the COS simulation is sensitive to" -> "To which parameters is the COS simulation sensitive"

Response: Corrected.

L84: "calculated" -> "calculates" (harmonize verb tenses.)

Response: Thanks for your comment. We have reviewed the verb tenses in the manuscript.

L111: "locations" -> "Locations" (capital letter L)

Response: Corrected.

L116: "three" -> "two" (I see only GLOBMAP and GLASS.)

Response: Thanks for your comment. We have revised the manuscript to include a more detailed description of the LAI data used in this study. "**The LAI dataset used here are the GLOBMAP global leaf area index product (Version 3) (see GLOBMAP global Leaf Area Index since 1981 | Zenodo) and the Global Land Surface Satellite (GLASS) LAI product (Version 3) (acquired from ftp://ftp.glcf.umd.edu/). They represent Leaf area index at a spatial resolution of 8 km (Liu et al., 2012) and 1 km (Xiao et al., 2016) respectively, and a temporal resolution of 8-day. With reference to the observed LAI at these sites (Wehr et al., 2017; Rastogi et al., 2018; Spielmann et al., 2019; Kohonen et al., 2022), we used GLOBMAP products to drive the BEPS model at most sites (5/7) duo to its good agreement with the observed LAI. Specifically, as the GLOBMAP product had considerably underestimated LAI at DK-Sor and was not consistent with the vegetation phenology at ES-Lma during the measurement campaign (Spielmann et al., 2019), GLASS LAI was used at these two sites. In addition, these LAI products were interpolated into daily values by the nearest neighbor method for the simulation.**" (line 160-168)

L163: "in in" -> "in"

Response: Corrected.

L168, 387: "In specific" -> "Specifically"

Response: Corrected.

L181: "if all model parameters is considered" -> "if all model parameters are considered"

Response: Corrected.

L184: "compensating with each other" -> "compensating each other"

Response: Corrected.

L193-194: "Taking into account the influence of the prior distribution to the behavioral parameter sets" -> "Taking into account the influence of the prior distribution of the behavioral parameter sets"

Response: Corrected.

L220 (twice): Fig. 1 -> Fig. 2

Response: Thanks for your comment. We have thoroughly reviewed the manuscript to ensure accurate referencing of figures and tables.

L225: "parameters related energy balance" -> "parameters related to energy balance'

Response: Corrected.

L281: "Fig. 2" -> "Fig. 3"

Response: Corrected.

L287: "e.g. Fig. 2d" -> "e.g., Fig. 3d"

Response: Corrected.

L297: "a further underestimate of the" -> "a further underestimation of the"

Response: Corrected.

L307: "the ensemble mean deviate remarkable from observations" -> "the ensemble mean strongly deviates from the observations"

Response: Corrected.

L320, 352: "posterior (green)" ->

Response: Thanks for your comment. We have rechecked the manuscript to avoid any errors in color representation.

L322, 354: "blue dots" -> -> It seems gray.

Response: Corrected.

L334: "Fig. 3" -> "Fig. 4"

Response: Corrected.

L363-364: "by influence the modeling of stomatal conductance" -> "by influencing the modeling of the stomatal conductance"

Response: Corrected.

L384: "in photosynthetic machinery" -> "in the photosynthetic machinery"

Response: Corrected.

L397: "confident levels" -> "confidence levels"

Response: Corrected.

L399: "A total of 14 parameter combinations demonstrate significantly correlated" -> "A total of 14 parameter combinations demonstrate significant correlations"

Response: Corrected.

L411, 596: The red font looks weird, like mixed with a black one, could the authors improve that?

Response: Of course. Now we have changed the font color and redrawn the figure.

L419: "e.g." -> "e.g.,"

Response: Corrected.

L450: "exhibits low sensitivity" -> "exhibits a low sensitivity"

Response: Corrected.

L468: "for COS simulation" -> "for COS simulations"

Response: Corrected.

L475: "such as that" -> "for example considering that"

Response: Thanks for your valuable comment. We have modified the sentence accordingly.

L478: "e.g." -> "e.g.,"

Response: Corrected.

L495: "interactions exists" -> "interactions exist"

Response: Corrected.

L495-496: "In particularly" -> "Particularly" or "In particular"

Response: Corrected.

L519: "according the" -> "according to the"

Response: Corrected.

L525: In the first exponential of equation (A7), "Kn" should be "kn".

Response: Corrected. Thank you for your detailed comment.

L530: "is the is the" -> "is the"

Response: Corrected.

L535: "($gs$w in)" -> The unit is missing.

Response: Thanks for your comment. Now the unit ($\mathrm{mol\,m^{-2}s^{-1}}$) of the leaf stomatal conductance for water vapor ($g_{sw}$) has been added.

L538: "is intercept" -> "is the intercept"

Response: Corrected.

L549: "the number of soil layer" -> "the number of soil layers"

Response: Corrected.

L572: In equation (A19), shouldn't "LAI" be "L" as in equations (A6/7)?

Response: Thanks for your valuable comment. According to Chen et al. (2012), here "L" actually denote the canopy depth (m), and we have corrected the error accordingly.

L590: "of the 9 parameters were" -> "of the 9 parameters that were"

Response: Corrected.

L591: "to the parameter dependent" -> "to the parameter dependency"

Response: Corrected.

L620: "Reference" -> "References"

Response: Corrected.

[revised manuscript text omitted]